# Inhibition of mutant EGFR in lung cancer cells triggers SOX2-FOXO6-dependent survival pathways

S Michael Rothenberg[1,2], Kyle Concannon[1], Sarah Cullen[1], Gaylor Boulay[1,3], Alexa B Turke[1], Anthony C Faber[1], Elizabeth L Lockerman[1], Miguel N Rivera[1,3], Jeffrey A Engelman[1,2], Shyamala Maheswaran[1,4], Daniel A Haber[1,2,5]*

[1]Cancer Center, Massachusetts General Hospital, Harvard Medical School, Charlestown, United States; [2]Department of Medicine, Massachusetts General Hospital, Harvard Medical School, Charlestown, United States; [3]Department of Pathology, Massachusetts General Hospital, Harvard Medical School, Boston, United States; [4]Department of Surgery, Massachusetts General Hospital, Harvard Medical School, Charlestown, United States; [5]Howard Hughes Medical Institute, Massachusetts General Hospital, Harvard Medical School, Charlestown, United States

**Abstract** Treatment of *EGFR*-mutant lung cancer with erlotinib results in dramatic tumor regression but it is invariably followed by drug resistance. In characterizing early transcriptional changes following drug treatment of mutant EGFR-addicted cells, we identified the stem cell transcriptional regulator SOX2 as being rapidly and specifically induced, both in vitro and in vivo. Suppression of SOX2 sensitizes cells to erlotinib-mediated apoptosis, ultimately decreasing the emergence of acquired resistance, whereas its ectopic expression reduces drug-induced cell death. We show that erlotinib relieves EGFR-dependent suppression of FOXO6, leading to its induction of SOX2, which in turn represses the pro-apoptotic BH3-only genes *BIM* and *BMF*. Together, these observations point to a physiological feedback mechanism that attenuates oncogene addiction-mediated cell death associated with the withdrawal of growth factor signaling and may therefore contribute to the development of resistance.

*For correspondence: dhaber@mgh.harvard.edu

## Introduction

The invariable development of drug resistance presents a critical challenge to the success of targeted cancer therapies (*Jänne et al., 2005*; *O'Hare et al., 2006*; *Poulikakos and Rosen, 2011*). Several mechanisms leading to such acquired resistance have been identified in patients with *EGFR*-mutant non-small cell lung cancer (NSCLC) treated with small molecule EGFR inhibitors such as erlotinib. Following dramatic initial tumor shrinkage, tumor regrowth is most frequently associated with the emergence of a secondary genetic change, the T790M 'gatekeeper' mutation within the EGFR kinase domain, which restores ATP binding in the presence of drug (*Pao et al., 2004*; *Kobayashi et al., 2005*; *Yun et al., 2007*). In other cases, amplification of related receptor tyrosine kinases (e.g., *MET*) or mutational activation of downstream kinases (e.g., *BRAF*, *PIK3CA*) may bypass the effect of EGFR inhibition (*Engelman et al., 2007*; *Sequist et al., 2011*; *Ohashi et al., 2012*). Different drug resistance mechanisms may coexist within different metastatic lesions of individual patients. Recently, clinical trials involving rebiopsy of tumor lesions at the earliest sign of drug resistance have also revealed phenotypic conversions that may contribute to drug resistance, including activation of epithelial-to-mesenchymal transition (EMT) and the remarkable trans-differentiation of lung cancers from

**eLife digest** Tumors can form when cells gain mutations in genes that enable them to grow and divide rapidly. In some human lung cancers, genetic mutations are found in a gene that makes a protein called EGFR. This protein encourages cells to divide and the mutations can lead to the cancer cells producing more EGFR, or producing a form of the protein that is more active.

Treating these cancers with a drug called erlotinib inhibits EGFR and makes the tumors shrink dramatically, but the tumors will usually re-grow because any tumor cells that survive often become resistant to the drug. There are several ways that the tumor cells can become resistant, which makes the task of developing a solution to this problem more difficult. It has been suggested that the tumor cells may enter a temporary 'drug-tolerant' state that helps them to survive and makes it more likely that they will develop resistance to the drug. However, it is not clear how this drug-tolerant state might work.

To address this question, Rothenberg et al. examined which genes are switched on (or 'expressed') in tumor cells with a mutant version of EGFR after they were treated with the erlotinib drug. The experiments show that a gene called *SOX2* is expressed in these cells. Cells that had lower levels of *SOX2* expression were more sensitive to the effects of the drug and fewer cells developed resistance. On the other hand, cells that had higher levels of *SOX2* expression were less sensitive to the drug and resistance was more likely to develop.

A protein called FOXO6—which is usually suppressed by EGFR—activates the *SOX2* gene in these cells. Therefore, using erlotinib to inhibit EGFR to kill the cancer cells increases the activity of FOXO6, which in turn promotes the survival of some of the cells by activating the *SOX2* gene. A better understanding of the ways in which cancer cells adapt to erlotinib and other drugs may help us to design more effective treatments with better outcomes for patients.

adenocarcinoma to small cell histologies (*Thomson et al., 2005*; *Witta et al., 2006*; *Sequist et al., 2011*). While some well-defined resistance mechanisms, such as the T790M-EGFR gatekeeper mutation and *MET* amplification, may be addressed using second line targeted drugs, the plasticity of cancer cell adaptation to disrupted oncogenic signaling poses a major challenge to the long-term success of these promising therapies.

In vitro modeling of acquired resistance to EGFR inhibitors has raised the possibility that a transient so-called 'drug-tolerant' state may precede the development of mutationally defined, heritable drug resistance (*Sharma et al., 2010*). By analogy with bacterial models of antibiotic resistance, such an intermediate state may be unstable, but enable treated cells to survive in the presence of drug long enough to acquire mutations that ultimately confer sustained drug resistance (*Balaban et al., 2004*). In PC9 mutant, EGFR-addicted lung cancer cells, EGFR inhibition triggers apoptosis in the vast majority of cells in vitro, uncovering approximately 0.3% that are drug tolerant, quiescent, and expressing the stem cell marker CD133 and the histone H3K4 demethylase KDM5A (*Sharma et al., 2010*). These drug tolerant cells readily revert to a drug-sensitive state following removal of the EGFR inhibitor, and their emergence in vitro is suppressed by treatment with an EGFR inhibitor combined with inhibitors of either histone deacetylases (HDACs) or the IGF-1 receptor. While this intermediate resistance mechanism remains to be validated in the clinical setting, it raises the possibility of suppressing pre-conditions that favor the acquisition of drug resistance, in order to circumvent the challenge of treating multiple established drug-resistant pathways.

Beyond the selection of cancer cell populations with transient drug-resistant phenotypes, recent studies of targeted cancer drugs have defined more rapid signaling feedback loops that modulate the cellular response to growth factor inhibition. For instance, acute loss of ERK signaling triggered by RAF or MEK inhibitors in *BRAF* mutant melanoma cells relieves ERK-dependent inhibition of RAS and CRAF, whose activation through ErbB receptor signaling may lead to paradoxical proliferative signals (*Pratilas et al., 2009*; *Paraiso et al., 2010*; *Lito et al., 2012*). Similarly, in *BRAF* mutant colorectal cancers, feedback activation of EGFR-dependent signaling attenuates the consequences of mutant BRAF inhibition, suppressing the apoptotic effect (*Corcoran et al., 2012*; *Prahallad et al., 2012*). In addition to signaling feedback loops, transcriptional outputs that generally limit cell proliferation have

also been implicated following disruption of EGFR activity, including the expression of transcriptional repressors, regulators of mRNA stability and microRNAs (*Kobayashi et al., 2006*; *Amit et al., 2007*; *Avraham et al., 2010*).

Here, we screened for early, unique transcriptional changes following erlotinib treatment in mutant EGFR-addicted cells, identifying highly specific induction of SOX2, a master transcriptional regulator required for embryonic stem cell maintenance. SOX2 represses the expression of pro-apoptotic molecules that mediate death following oncogene withdrawal in these cells. The induction of SOX2 results from the activation of FOXO6, a forkhead family transcription factor, following EGFR inhibition. Knockdown or ectopic expression of SOX2 modulates the degree of apoptosis observed following oncogene withdrawal and promotes drug resistance, pointing to a novel homeostatic mechanism that may contribute to cellular adaptation to the withdrawal of growth factor signaling, which underlies most approaches to targeted cancer therapy.

## Results

### SOX2 is specifically induced in *EGFR*-mutated lung cancer cells following treatment with the EGFR inhibitor erlotinib

To interrogate the transcriptional response to EGFR inhibition, we used HCC827 lung cancer cells, harboring an amplified mutated *EGFR* allele (in-frame deletion of 15 nucleotides in exon 19) and displaying exquisite sensitivity to the EGFR inhibitor erlotinib. Cell cultures were treated in triplicate with 1 μM erlotinib for 6 hr, followed by mRNA isolation and whole transcriptome analysis (Affymetrix U133 Plus 2.0 expression arrays) (*Rothenberg, 2015*). A total of 35 genes showed >fourfold change in expression (FDR <0.05), including 22 downregulated and 13 upregulated transcripts (represented by 48 unique probe sets; *Figure 1—figure supplement 1A*). Among induced transcripts, SOX2 was unique in the specificity and rapidity of its induction following EGFR inhibition (*Figure 1*, *Figure 1—figure supplement 1B*). Thus, SOX2 was strongly induced in three mutant EGFR-addicted lung cancer cell lines (HCC827, PC9, H3255) following treatment with physiologically relevant concentrations of erlotinib (0.1 μM), but not when these cells were treated with comparably effective doses of cytotoxic chemotherapy (*Figure 1A,B* and *Figure 1—figure supplement 2A*). SOX2 was also not induced in other oncogene-dependent models, such as *ALK*-translocated lung cancer cells treated with crizotinib, *HER2*-amplified breast cancer cells exposed to lapatinib or *BRAF*-mutant melanoma cells treated with AZD6244 (*Figure 1A* and *Figure 1—figure supplement 2B*). Consistent with its dependence on suppression of mutant EGFR signaling in the context of EGFR 'addiction', SOX2 was not induced following erlotinib treatment of H1975 cells, which harbor both an EGFR activating mutation and the T790M gatekeeper mutation that confers resistant to erlotinib; or in H1650 cells with mutated EGFR that are relatively resistant to the effects of EGFR inhibition in part through genetic loss of *PTEN* (*Figure 1—figure supplement 2B*) (*Sos et al., 2009*). However, treatment of H1975 cells with the L858R/T790M mutation-selective inhibitor WZ4002 resulted in SOX2 induction (*Figure 1—figure supplement 2B*, right) (*Zhou et al., 2009*). In cells that show erlotinib-mediated induction of SOX2, siRNA-mediated knockdown of EGFR also led to strong induction of SOX2 (in the absence of erlotinib), confirming the specificity of the drug effect (*Figure 1C*). Simultaneous treatment of cells with actinomycin D and erlotinib suppressed the induction of SOX2, consistent with a primary effect of EGFR inhibition in increasing SOX2 transcript levels (*Figure 1—figure supplement 2C*).

Other transcripts induced or repressed following erlotinib treatment of mutant EGFR-addicted cells were not selective to EGFR signaling. Downregulated genes included known direct transcriptional targets of ERK signaling (*CCND1, FOSL1, EGR1, IER3, IL-8*) and shared feedback inhibitors of receptor tyrosine kinase (RTK) signaling (*DUSP6*) (*Amit et al., 2007*). This gene set overlaps with genes known to be differentially expressed following treatment of *BRAF*-mutant melanoma cells with a MEK inhibitor and exposure of *EGFR*-mutant lung cancer cells to an irreversible EGFR inhibitor (*Figure 1—figure supplement 3*) (*Kobayashi et al., 2006*; *Pratilas et al., 2009*). In addition to SOX2, the 12 other transcripts induced by EGFR inhibition included genes encoding metabolizing enzymes (*CYP1B1, CYP1A1*) that are normally induced by treatment with a variety of small chemical entities, genes that we also found to be equally well induced by inhibition of downstream signaling pathways (MEK/MAPK inhibition using AZD6244 and PI3K/mTOR inhibition

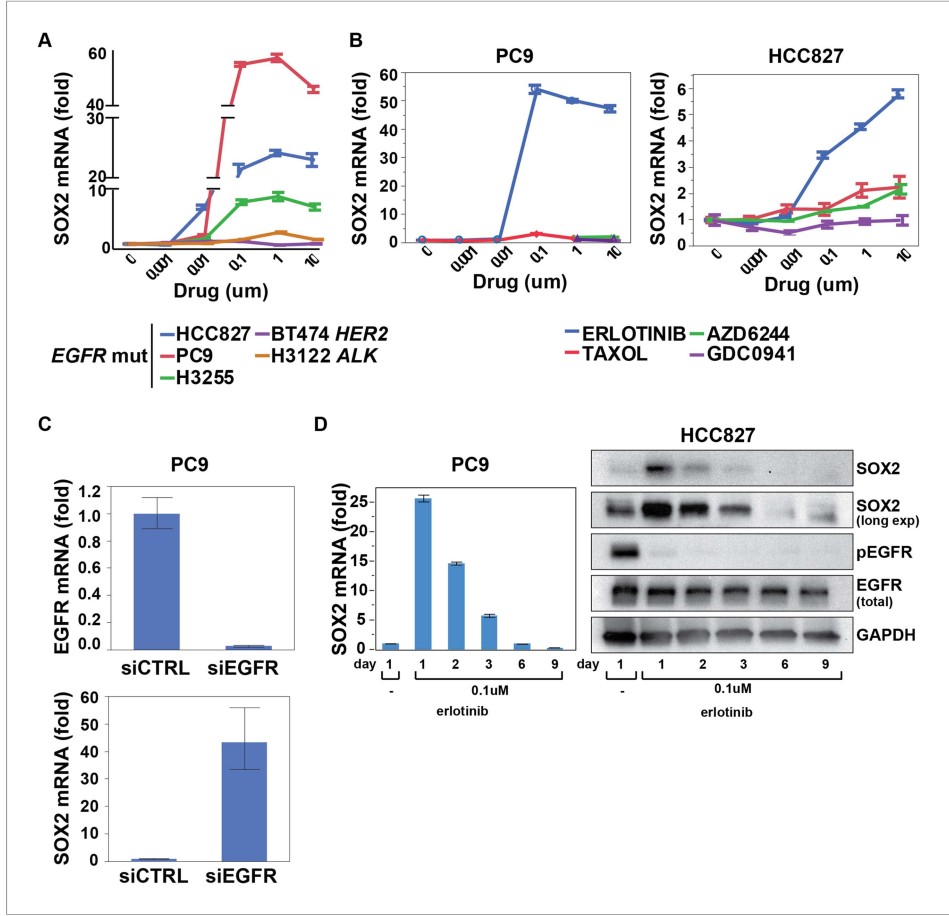

**Figure 1**. SOX2 transcript is specifically induced by erlotinib in EGFR-mutant and addicted lung cancer cell lines. (**A**) Cell lines were treated with an inhibitor of the driving oncogenic lesion for 24 hr (erlotinib for *EGFR*-mutant, lapatinib for *HER2*-amplified and crizotinib for *ALK*-translocated cells), followed by isolation of total RNA and quantitative PCR for SOX2 transcript. (**B**) PC9 and HCC827 cells were treated with different agents, followed by quantitative PCR for SOX2. The IC50 for PC9 of erlotinib, taxol, AZD6244, and GDC0941 is 0.05, 0.005, 5, and >10 µM; for HCC827, 0.1, 0.01, >10, and 1 µM (data not shown). (**C**) PC9 cells were transfected with control siRNA or siRNA targeting EGFR. 48 hr after transfection, the levels of SOX2 and EGFR were determined by qPCR. (**D**) PC9 and HCC827 cells were treated continuously with 0.1 µM erlotinib for 9 days, with fresh media/drug added every 3 days. SOX2 level at each time point was analyzed by qPCR (left) or immunoblot (right). All qPCR data are displayed as mean Ct value (normalized to GAPDH and untreated cells) of 3–6 replicates −/+ SEM, with data in (**C**) normalized to untreated siCTRL cells.

The following figure supplements are available for figure 1:

**Figure supplement 1**. Gene expression profiling after erlotinib treatment.

**Figure supplement 2**. Effect of various treatments on SOX2 expression in different cell contexts.

**Figure supplement 3**. Overlap of differentially expressed genes.

**Figure supplement 4**. Time course of SOX2 induction by quantitative immunofluorescence microscopy.

---

using BEZ235), one transcript (*CCNG2*) previously described in another EGFR model (*Kobayashi et al., 2006*) and a long noncoding RNA (*NEAT1*). Taken all together, these results indicate that suppression of EGFR signaling in mutant EGFR-addicted lung cancer cells is highly specific in triggering transcriptional induction of SOX2.

## Induction of SOX2 in erlotinib-treated cells

SOX2 encodes a master transcriptional regulator, implicated in stem cell maintenance and iPS cell generation. It is also required for upper aerodigestive tract development, and is known to be amplified in a subset of esophageal and squamous lung cancers, although it has not been previously implicated in lung adenocarcinomas, including the subset driven by mutant EGFR (*Ellis et al., 2004*; *Gontan et al., 2008*; *Bass et al., 2009*). Remarkably, SOX2 expression following exposure of HCC827 cells to erlotinib was transient, peaking at 24 hr after exposure to therapeutic levels of the drug (*Figure 1D*). Thereafter, SOX2 expression returned to basal levels despite continued erlotinib treatment in surviving cells (*Figure 1D*, *Figure 1—figure supplement 4*).

The level of SOX2 induction in cultured cells exposed to erlotinib showed considerable heterogeneity, with a subset of cells (~20%, with some experimental variability) expressing high levels (*Figure 2A*). The SOX2+ fraction was not increased by higher drug dosage, beyond that required for full inhibition of EGFR (*Figure 2—figure supplement 1*). Given the link between SOX2 expression and cellular reprogramming, we first asked whether cells with the high SOX2 expression represent a subset with stem cell markers. However, SOX2 expression did not correlate with expression of the putative stem cell markers CD133, CD44, CD24, OCT-4, or KLF-4 (*Figure 2—figure supplement 2*) nor did microarray-based expression profiling of high SOX2-sorted cells identify a stem-like signature (data not shown). Nonetheless, SOX2-expressing cells had a very low proliferative index, as measured by Ki67 staining (0.5% of Ki67+/SOX2+ vs 51% Ki67+/SOX2− HCC827 cells at baseline [p = 0.015]; and 0.15% Ki67+/SOX2+ vs 6.4% Ki67+/SOX2− cells following erlotinib [p < 0.0001]) (*Figure 2C*, *Figure 2—figure supplement 3*).

To test whether the heterogeneous induction of SOX2 following EGFR inhibition represents a stochastic event or a heritable property shared by a subset of the parental population, we treated cells sequentially with pulses of erlotinib. Retreatment of cells after a period of recovery produced similar heterogeneity of SOX2+ cells as the initial treatment, pointing to the absence of enrichment for highly inducible cells (*Figure 2B* and *Figure 2—figure supplement 4A*). In addition, SOX2 could still be induced in cells made resistant to erlotinib, but only at the much higher doses of drug required to fully inhibit EGFR in resistant cells (*Figure 2—figure supplement 4B*). Finally, cloning of 5–6 individual HCC827 and PC9 cells consistently generated mixed populations, including high level SOX2 inducers together with non-expressing cells, demonstrating that this heterogeneity is likely stochastic, rather than heritable (*Figure 2C* and *Figure 2—figure supplement 5*). Thus, erlotinib treatment of *EGFR*-mutant cells results in transient and heterogeneous induction of SOX2, with a stochastic distribution, integrally tied to inhibition of EGFR, in which the cells with the highest expression have a low proliferative index.

## Induction of SOX2 in EGFR-mutant tumors following erlotinib treatment

To test the physiological significance of SOX2 induction following withdrawal of mutant EGFR signaling, we first made use of mouse tumor models. The effectiveness of EGFR inhibitors in treating patients with EGFR-mutant NSCLC is well modeled in mouse xenograft assays, where oral administration of erlotinib for a few days is sufficient to cause massive regression of established tumors. We generated PC9 cell-derived subcutaneous tumors in nude mice and treated these with a single oral dose of 100 mg/kg erlotinib when the tumors had reached approximately 500 mm³, harvesting tumors 24 hr after treatment. Immunohistochemical (IHC) analysis revealed minimal SOX2 expression in mock-treated xenografts (mean 0.2 SOX2+ nuclei per field), but clearly increased (and heterogeneous) SOX2 positive cells after a single dose of erlotinib (mean: 7.4 SOX2+ nuclei per field, N = 147–151 fields, p < 0.0001) (*Figure 3*). Similar studies in HCC827 cell-derived xenografts revealed low (but detectable) levels of SOX2 expression in mock-treated tumors; again, a single oral dose of erlotinib increased both the number of SOX2-positive cells and the level of SOX2 expression per nucleus (*Figure 3—figure supplement 1*). Thus, in a physiological setting that mimics the initial therapeutic response to EGFR inhibitors in *EGFR*-mutant NSCLC, treated cancer cells rapidly induce SOX2.

## Induction of SOX2 in patient-derived EGFR-mutant tumor cells

While established cancer cell lines, such as PC9 and HCC827, recapitulate the phenomenon of oncogene addiction, patient-derived cell lines directly cultured from biopsies may be more

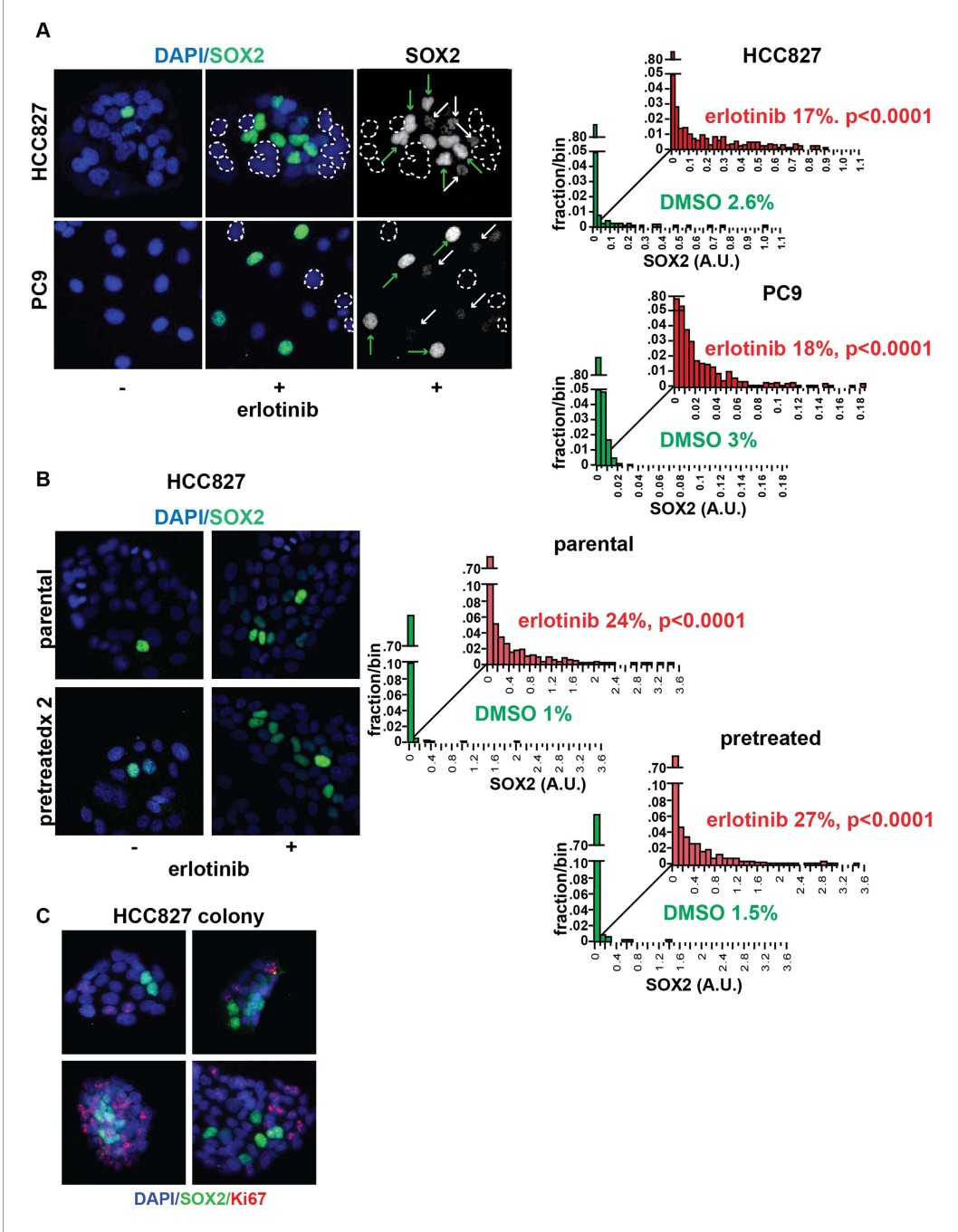

**Figure 2**. Induction of SOX2 in erlotinib-treated cells. (**A**) Left, HCC827 (upper) or PC9 (lower) cells were treated with 0.1 µM erlotinib for 24 hr, followed by immunofluorescence staining using an antibody to SOX2 and DAPI. For erlotinib-treated cells (middle and right pairs of images), the heterogeneity in induced SOX2 levels per cell in each population is indicated by dashed outlines indicating DAPI+ nuclei lacking SOX2, white arrows for nuclei with low (but detectable) SOX2 and green arrows for nuclei with high SOX2. Right, distribution of SOX2 fluorescence in each sample. Mean fluorescence counts for each cell were quantitated and normalized to exposure time as described in 'Materials and methods'. p < 0.0001 for the comparison of erlotinib-treated cells to DMSO (Student's *t*-test, unequal variances, N = 1219–3485, means are 0.005/0.04 for untreated/treated HCC827 and 0.001/0.008 for untreated/ treated PC9, % SOX2+ is shown). Source data are included as *Figure 2—source data 1, 2*. (**B**) Induction of SOX2 in erlotinib-retreated cells. HCC827 cells were treated with 1.0 µM erlotinib for 24 hr (75% cell killing), followed by removal of drug, replating of cells after 7 days of recovery and retreatment with the same concentration of erlotinib. This protocol was repeated, and then cells were treated a third time with erlotinib for 24 hr and analyzed by

*Figure 2. continued on next page*

*Figure 2. Continued*

immunofluorescence microscopy using antibodies to SOX2 and DAPI. The increase in SOX2-positive cells was highly significant for both erlotinib pretreated and untreated cells, but no enrichment was observed as a consequence of pretreatment p < 0.0001 for the comparison of erlotinib-treated cells to DMSO (Student's *t*-test, unequal variances, N = 1106–2143, means are 0.005/0.17 for untreated/treated-parental, 0.007/0.2 for untreated/treated-pretreated, % SOX2+ is shown). Source data are included as *Figure 2—source data 3*. (**C**) Immunofluorescence analysis of single colonies formed from single clones of HCC827 cells stained for DAPI (blue), SOX2 (green), and Ki67 (red).

The following source data and figure supplements are available for figure 2:

**Source data 1**. Raw immunofluorescence data for quantitation of SOX2 staining in HCC827 cells with erlotinib treatment in *Figure 2A*, and SOX2+ Ki67 staining in *Figure 2—figure supplement 3*.

**Source data 2**. Raw immunofluorescence data for quantitation of SOX2 staining in PC9 cells with erlotinib treatment in *Figure 2A* and *Figure 1—figure supplement 4*.

**Source data 3**. Raw immunofluorescence data for quantitation of SOX2 staining in HCC827 cells recovered after retreatment (x2) with erlotinib, compared to previously untreated, in *Figure 2B*.

**Source data 4**. Raw immunofluorescence data for quantitation of SOX2 staining in HCC827 and PC9 cells with increasing dose of erlotinib in *Figure 2—figure supplement 1*.

**Source data 5**. Raw immunofluorescence data for quantitation of SOX2 staining in PC9 cells recovered after retreatment (x2) with erlotinib, compared to previously untreated cells, in *Figure 2—figure supplement 4A*.

**Source data 6**. Raw immunofluorescence data for quantitation of phospho-EGFR (pY1068) in parental and erlotinib-resistant PC9 cells in *Figure 2—figure supplement 4B*.

**Figure supplement 1**. Increasing the dose of erlotinib does not significantly increase the fraction of SOX2+ cells.

**Figure supplement 2**. Stem cell markers do not colocalize with SOX2+ cells.

**Figure supplement 3**. SOX2 is expressed most highly in nonproliferative cells.

**Figure supplement 4**. Stochastic induction of SOX2 by erlotinib in PC9 cells.

**Figure supplement 5**. The highest induction of SOX2 in individually isolated subclones of *EGFR*-mutant cells occurs in a subset of cells, as for the parental cells.

**Figure supplement 6**. KDM5A is not induced following treatment of PC9 cells with erlotinib for 24 hr.

representative of heterogeneous primary cultures (*Crystal et al., 2014*). Such biopsies are typically obtained at the time of disease progression, where defining drug resistance mechanisms may shape further therapy. We therefore analyzed short-term cultures of EGFR-mutant cells derived by re-biopsy of two patients who had initially responded to erlotinib therapy but subsequently developed progressive disease due to the acquisition of a T790M gatekeeper mutation. SOX2 induction was absent following treatment with erlotinib, which was ineffective in inhibiting EGFR in these resistant patient-derived cells (*Figure 4*). However, the novel 'third line' irreversible, EGFR-mutant-specific inhibitor WZ4002 demonstrated potent EGFR inhibition in these cells, along with induction of SOX2 (*Figure 4*). Thus, SOX2 induction is consistently observed following acute withdrawal of EGFR signals in cancer models as well as in patient-derived cells that exhibit oncogene dependence on the EGFR pathway.

## Regulation of anti-apoptotic signals by SOX2

To examine the consequence of ectopic SOX2 expression, we generated HCC827 cells with a lentiviral-driven, doxycycline-inducible construct. Careful titration of doxycycline-permitted

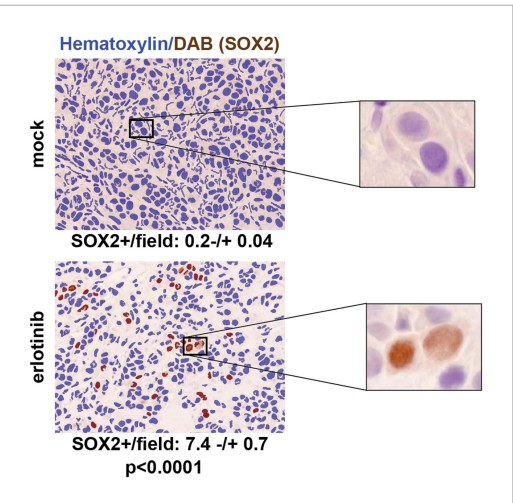

**Hematoxylin/DAB (SOX2)**

mock

SOX2+/field: 0.2-/+ 0.04

erlotinib

SOX2+/field: 7.4 -/+ 0.7
p<0.0001

**Figure 3**. SOX2 is induced by erlotinib in a subset of EGFR-mutant cells in vivo. Nude mice were xenografted subcutaneously with PC9 cells and treated with a single oral dose of erlotinib (100 mg/kg) (or carrier) when the tumors had reached ~500 mm³. Tumors were harvested 24 hr after treatment, and immunohistochemistry for SOX2 was carried out on formalin-fixed, paraffin embedded tumor specimens. The panels show automated scoring of SOX2+ nuclei (brown) as described in 'Materials and methods'. The actual IHC images for the areas indicated by rectangles are shown magnified to the right. p < 0.0001 for the comparison erlotinib-treated vs control (Student's *t*-test, unequal variances, N = 147–151 fields from 4 xenografts, mean SOX2+ nuclei/10× field −/+ SEM is shown for each treatment). Source data are included as *Figure 3—source data 1*.

The following source data and figure supplements are available for figure 3:

**Source data 1**. Number of SOX2+cells per field for quantitation of SOX2 staining in PC9 cell xenografts in *Figure 3*.

**Source data 2**. Raw absorbance data for quantitation of SOX2 staining in HCC827 cell xenografts in *Figure 3—figure supplement 1*.

**Figure supplement 1**. Erlotinib treatment results in induction of SOX2 in vivo.

induction of ectopic SOX2 to physiologic levels in the absence of erlotinib, comparable at the single cell level to endogenous SOX2 induction in the presence of erlotinib, though in a larger fraction of cells (*Figure 5A* and *Figure 5—figure supplement 1A*). As expected, treatment of control HCC827 cells with erlotinib led to the inhibition of EGFR signaling, as measured by decreased phosphorylation of EGFR, AKT, and ERK, and caused dramatic apoptosis, as determined by increased PARP and caspase-3 cleavage (*Figure 5A*, lanes 1–2). In contrast, expression of ectopic SOX2 significantly decreased erlotinib-mediated apoptosis, resulting in decreased PARP and caspase-3 cleavage (*Figure 5A*, lane 2 vs 4). Exogenous SOX2 itself did not alter basal or erlotinib-inhibited phosphorylation of EGFR, AKT, or ERK (*Figure 5A*, lanes 3–4).

To determine the functional significance of endogenous SOX2 induction, we next used siRNAs to block its induction in erlotinib-treated cells. While both HCC827 and PC9 cells are highly sensitive to EGFR inhibition at baseline, SOX2 knockdown further increased erlotinib-induced apoptosis, as determined by PARP and caspase-3 cleavage assays and by cell enumeration (*Figure 5B,D* and *Figure 5—figure supplement 2*). The apoptotic effect of the most potent siRNA was rescued by expression of an ectopic siRNA-resistant SOX2 construct (*Figure 5C*) and individual siRNAs-induced apoptosis in proportion to the degree of knockdown (*Figure 5—figure supplement 3*), confirming the specificity of the effect. As with expression of exogenous SOX2, endogenous SOX2 suppression itself did not have a consistent effect on EGFR signaling, as measured by phosphorylation of EGFR, AKT, or ERK (*Figure 5B*).

Given the effect of transient SOX2 induction in acutely modulating cell survival following erlotinib treatment of EGFR-mutant cells, we tested whether this translates into a longer term impact on acquired drug resistance. We used the well-characterized PC9 cell model of EGFR-mutant NSCLC to transfect siRNAs against SOX2 (the same duplex validated by rescue in *Figure 5C*) or control, followed by continuous exposure to 1.0 μM erlotinib (*Figure 5E*). As expected, SOX2 knockdown alone had no significant toxicity, while erlotinib treatment led to massive cell death at 3 days (*Figure 5E*, left and middle images). At high magnification, many more individual surviving cells remained in control transfected cells than after transfection with siRNA targeting SOX2 (*Figure 5E*, middle insets). With continued incubation in erlotinib, drug-resistant colonies emerged in control transfected cultures at higher frequency than in those treated with siRNA targeting SOX2 (*Figure 5E*, right). Consistent with the short-term duration of siRNA effectiveness, these results suggest that preventing the short-term induction of SOX2 allows fewer cells to survive the initial exposure to

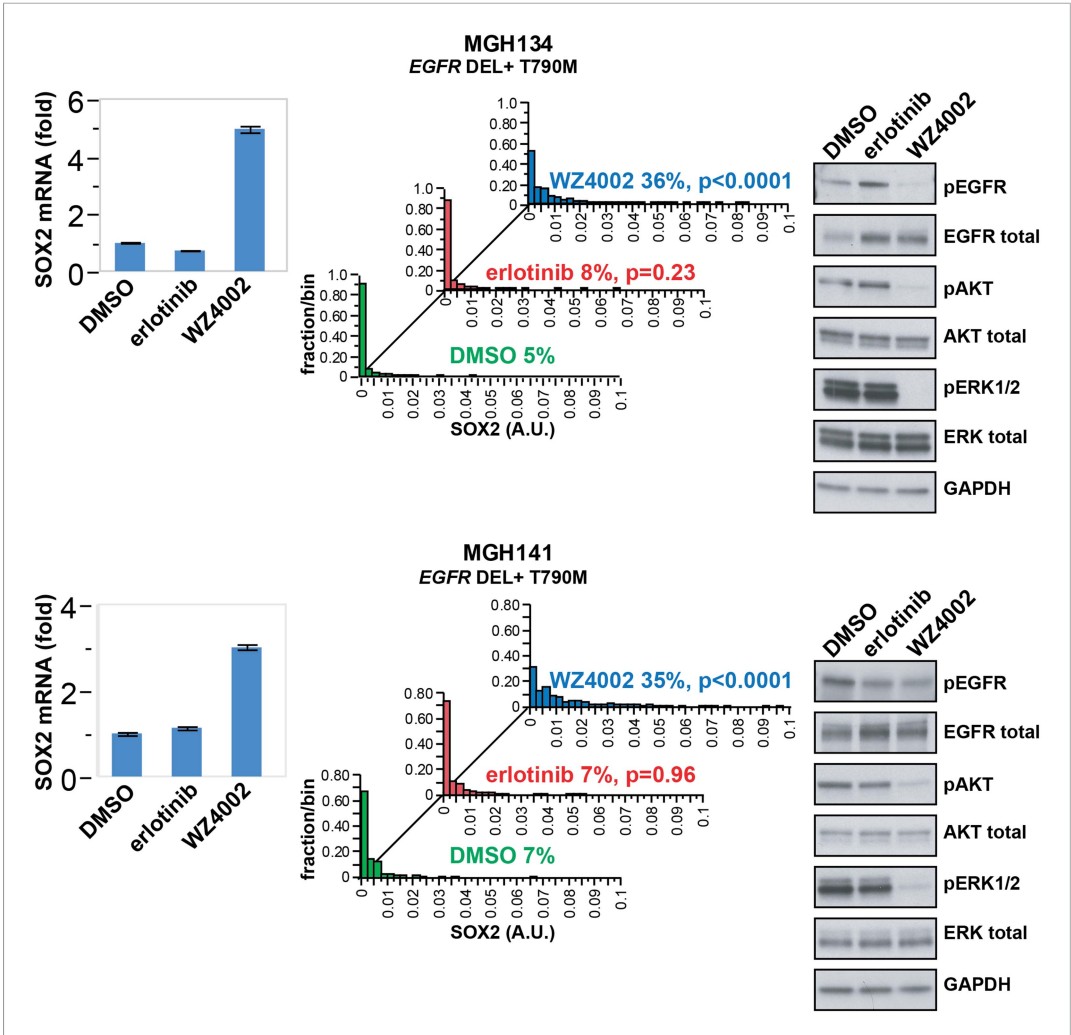

**Figure 4**. SOX2 is induced by therapy targeting the resistance genotype in cell lines derived by rebiopsy of patients. Short-term cultures of tumor cells derived from patients at the time of acquired resistance (both tumors *EGFR* genotype exon 19 deletion + T790M) were treated with the indicated agents for 24 hr, followed by isolation of total RNA and qPCR for SOX2 transcript (left panels) or quantitative immunofluorescence analysis after staining with antibodies to SOX2 (middle panels). The effect of each treatment on downstream signaling was determined by immunoblot analysis with the indicated antibodies (right panels). For qPCR, data are displayed as the mean of 4 replicates −/+ SEM. For histograms, p-values are shown for the comparison of each treatment to DMSO (Student's *t*-test, unequal variances, N = 229–1808, means are 0.001/0.0014/0.0054 for DMSO/erlotinib/WZ4002-treated MGH134 and 0.003/0.003/0.011 for DMSO/erlotinib/WZ4002-treated MGH141, % SOX2+ is shown). Source data are included as *Figure 4—source data 1*.

The following source data are available for figure 4:

**Source data 1**. Raw immunofluorescence data for quantitation of SOX2 staining with different treatments in patient-derived tumor cells.

erlotinib, preventing an adaptation response that ultimately delays the emergence of erlotinib-resistant colonies.

## Erlotinib-induced SOX2 directly regulates expression of *BIM* and *BMF*

The mechanisms underlying the cell death response of cancer cells following withdrawal of oncogene-addicting signals are complex but appear to result in part from activation of the pro-apoptotic

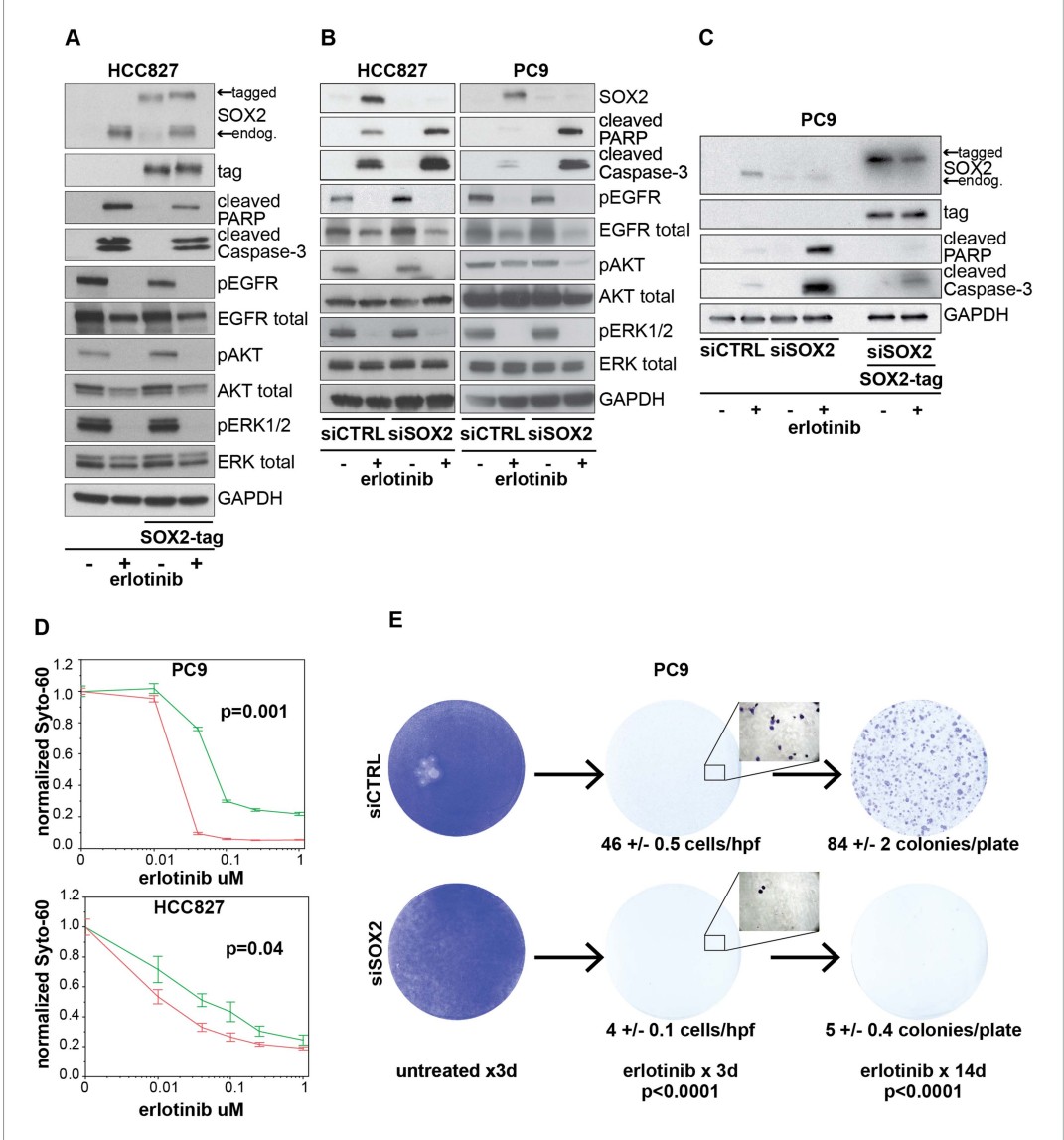

**Figure 5**. Induction of SOX2 protects cells from erlotinib-induced apoptosis. (**A**) HCC827 cells were stably transduced with a doxycycline inducible epitope-tagged SOX2 lentiviral expression vector. Doxycycline was added for 3 hr ('SOX2-tag') and then removed prior to the addition of DMSO or 0.1 μM erlotinib for 24 hr, followed by immunoblot of protein lysates with the indicated antibodies. Exogenous SOX2 migrates more slowly than the endogenous protein due to the presence of the tag. (**B**) HCC827 (left) or PC9 (right) cells were transfected with control siRNA or siRNA targeting SOX2. 24-hr after transfection, DMSO or 0.1 μM erlotinib was added. The effect of SOX2 knockdown was assessed by immunoblot analysis of protein lysates with the indicated antibodies after overnight treatment. (**C**) PC9 cells were stably transduced with a tagged SOX2 lentiviral vector in which silent mutations were introduced into the target site for the most potent siRNA against SOX2. Cells were transfected with the indicated siRNAs, treated with doxycycline followed by erlotinib as in (**A**), and protein lysates were analyzed by immunoblot with the indicated antibodies after overnight treatment. The increased PARP and caspase-3 cleavage observed when erlotinib treatment is combined with siRNA targeting SOX2 (lane 4) is suppressed by siRNA-resistant, exogenous SOX2 (lane 6). (**D**) PC9 and HCC827 cells were transfected in 96-well plates and treated 24 hr later with a dilution series of erlotinib, followed by Syto-60 assay. Data are displayed as the mean of 3–5 replicates −/+ SEM. p = 0.001 (PC9) and 0.04 (HCC827) for the comparison of mean IC50 for siCTRL vs siSOX2 (Student's *t*-test, unequal variances). (**E**) Preventing SOX2 induction using siRNA decreases the development of acquired erlotinib resistance. PC9 cells were transfected with control siRNA or siRNA targeting SOX2, followed by treatment after 24 hr with 1.0 μM erlotinib. Erlotinib-containing medium was renewed every 3 days, and plates were fixed and stained with Crystal Violet at the indicated times. The left panels demonstrate the absence of toxicity following transfection with

*Figure 5. continued on next page*

*Figure 5. Continued*

siRNA targeting SOX2 in the absence of erlotinib. Middle panels demonstrate cell loss after 3 days of treatment, due to erlotinib-induced apoptosis; at higher magnification, more control cells remain attached than cells transfected with siRNA against SOX2. Right panels show colonies of proliferating cells after 2 weeks of continuous erlotinib treatment. p < 0.0001 for the number of cells per 20× field (N = 33 fields per sample) or the number of colonies per plate (N = 9 plates per sample from three independent experiments), for siRNA targeting SOX2 vs control cells (Student's *t*-test, unequal variances).

The following source data and figure supplements are available for figure 5:

**Source data 1**. Raw immunofluorescence data for quantitation of SOX2 staining in HCC827 cells with inducible SOX2 in *Figure 5—figure supplement 1A*.

**Source data 2**. Raw immunofluorescence data for quantitation of SOX2 and cleaved caspase-3 costaining in PC9 cells transfected with siCTRL or siSOX2 in *Figure 5—figure supplement 2*.

**Figure supplement 1**. Induction of exogenous SOX2.

**Figure supplement 2**. SOX2 expression modulates erlotinib-induced apoptosis.

**Figure supplement 3**. The effect of siRNA targeting SOX2 is specific.

**Figure supplement 4**. Quantitation of the effect of SOX2 knockdown on BIM levels.

proteins BIM and PUMA (*Costa et al., 2007*; *Gong et al., 2007*; *Bean et al., 2013*). To search for additional targets that mediate SOX2's anti-apoptotic effect, we screened for altered expression of multiple pro-apoptotic and anti-apoptotic proteins in cells with overexpression of ectopic SOX2, following treatment with erlotinib. In both HCC827 and PC9 EGFR-mutant lung cancer cells, three of 14 BH3-domain containing proteins tested showed strongly reduced mRNA induction by erlotinib in the presence of ectopic SOX2: BIM, BMF, and HRK (but not PUMA) (*Figure 6A* and *Figure 6—figure supplement 1*). All three of these are known to induce apoptosis to different degrees, depending on the type of apoptotic stimulus. The effect of ectopic SOX2 on BIM protein levels was confirmed using two different expression strategies (*Figure 5—figure supplement 1B,C*). Ectopic expression of SOX2, both at high and physiological levels, attenuated the erlotinib-induced induction of SOX2. In contrast, SOX2 knockdown coincident with erlotinib treatment led to an overall increase in BIM levels, although this effect was attenuated when averaged across a cell population with heterogeneous induction of endogenous SOX2 (*Figure 5—figure supplement 4*).

To determine whether BIM and BMF are endogenous SOX2 target genes, we carried out chromatin immunoprecipitation sequencing (ChIP seq) experiments, using erlotinib-treated HCC827 cells. Strong SOX2-binding peaks were present within ~2 kB of the transcriptional start sites (TSS) of both *BIM* and *BMF* genes (for BMF, the peak spans the TSS; for *BIM*, it is located within the first intron) (*Figure 6B*, upper). Importantly, ChIP qPCR demonstrated significant enhancement of SOX2 binding with erlotinib treatment at both peaks compared to untreated cells, suggesting that binding is functionally significant (*Figure 6B*, lower).

Consistent with their functional roles, knockdown of either BIM or BMF (but not HRK) decreased erlotinib-induced apoptosis, with combined BIM and BMF knockdown displaying an additive effect (*Figure 6C*). The role of BMF could be even more important, since BMF knockdown increases BIM levels, which may blunt the anti-apoptotic effect of BMF loss (*Figure 6C*, left). Together, these results suggest that SOX2 induction following erlotinib exposure in EGFR-mutant cells suppresses transcriptional induction of the BH3-only *BIM* and *BMF* genes, which contribute to apoptosis following oncogene withdrawal.

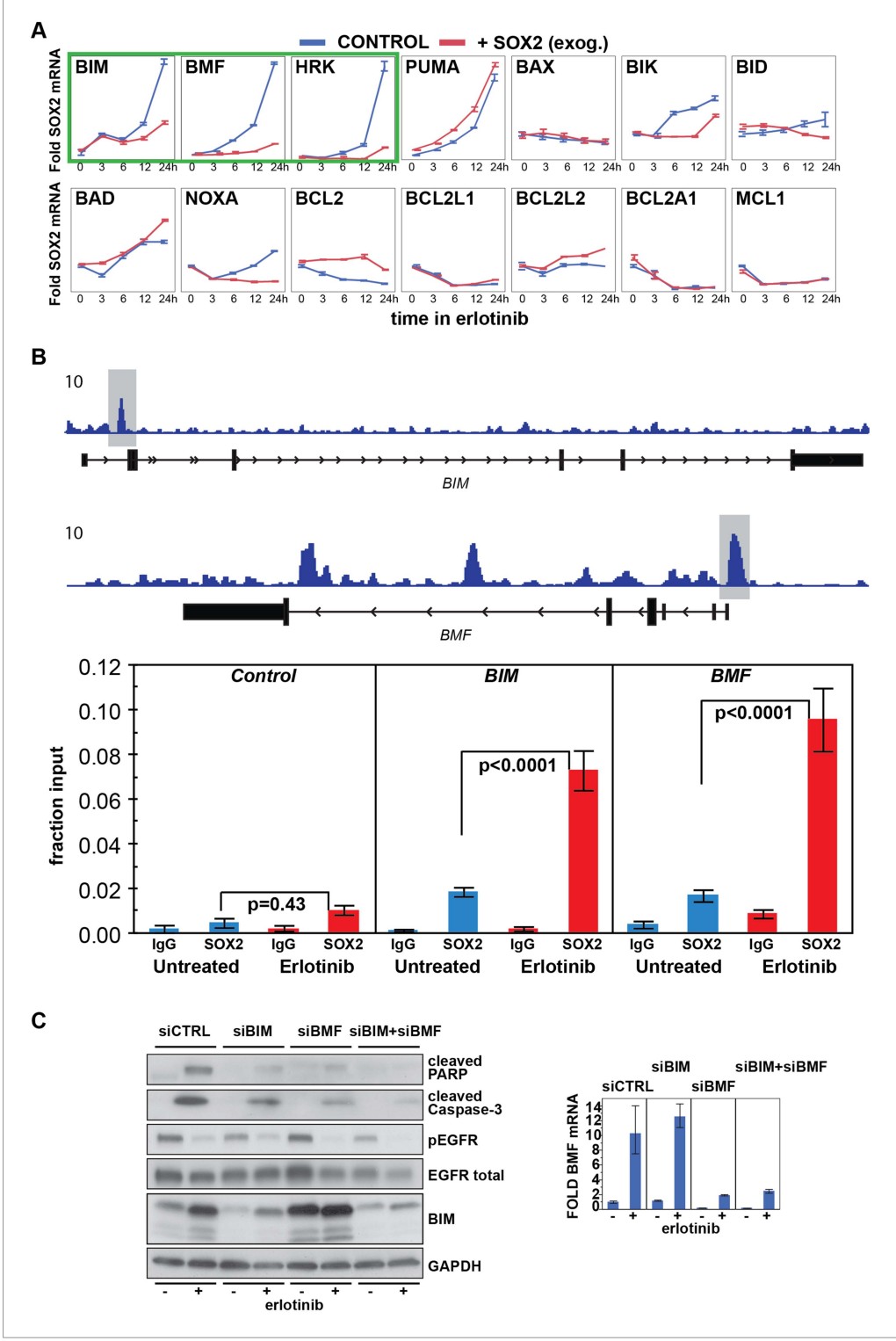

**Figure 6**. Erlotinib-induced SOX2 directly regulates expression of *BIM* and *BMF*. (**A**) The levels of transcripts for each of the indicated BH3 domain-containing proteins was assessed by quantitative PCR at multiple time points after erlotinib treatment in uninduced PC9 cells (blue lines) and in cells in which expression of SOX2 was induced with doxycycline (red lines), which was not removed prior to erlotinib addition in order to further increase SOX2 levels, as shown in *Figure 5—figure supplement 1B*. The y-axis maximum for all graphs is set to 4 except for HRK (y maximum = 11) and BMF (y maximum = 110). Data are displayed as mean Ct of 4 replicates (normalized to untreated cells and GAPDH) −/+ SEM. (**B**) Upper panel, ChIP seq demonstrates SOX2 binding to *BIM* and *BMF*. HCC827 cells
*Figure 6. continued on next page*

*Figure 6. Continued*

were treated with 0.1 μM erlotinib for 24 hr, followed by chromatin immunoprecipitation (ChIP) using anti-SOX2 antibody and ChIP Seq as described in 'Materials and methods'. ChIP seq signal tracks are displayed. Lower panel, HCC827 cells were left untreated or were treated with 0.1 μM erlotinib for 24 hr, followed by chromatin immunoprecipitation using anti-SOX2 antibody or IgG as a negative control, and qPCR with control primers or primers within the peaks indicated by the gray boxes in the ChIP seq tracks. Data are displayed as mean Ct value (normalized to input chromatin) of 4 replicates –/+ standard error, p-values for the comparison of untreated vs erlotinib treated cells are shown (Student's *t*-test, unequal variances). (**C**) Knockdown of BIM and BMF decreases apoptosis. Left panel, PC9 cells were transfected with siRNA constructs targeting BIM and BMF (alone or together) or a control siRNA. 24 hr after transfection, cells were treated with DMSO or 0.1 μM erlotinib. 24 hr after treatment, protein lysates were prepared, and immunoblot was performed with the indicated antibodies. Right panel, the efficiency of knockdown of BMF was confirmed by qPCR.

The following figure supplement is available for figure 6:

**Figure supplement 1**. Effect of SOX2 overexpression on apoptotic regulators.

## Induction of SOX2 following EGFR inhibition is regulated by FOXO6

To search for mediators of SOX2 induction, we explored the Molecular Signatures and TRANSFAC databases for transcription factor target sequences within the promoters of the 12 highest erlotinib-induced genes (*Wingender et al., 2000*; *Subramanian et al., 2005*). Several binding motifs for FOXO proteins were highly significantly enriched (q-value = 0.003 or less): for SOX2, multiple sites were present within 2 kb of the transcriptional start site (*Figure 7A* and *Figure 7—figure supplement 1*). Expression of all of the FOXO family members was detectable at baseline in HCC827 cells and erlotinib treatment (8 hr) was associated with a 1.6–4.4-fold induction (*Figure 7B*), as well as with loss of the AKT-mediated inhibitory N-terminal threonine phosphorylation of the FOXO proteins (*Figure 7—figure supplement 2A*).

Given the evidence of erlotinib-mediated FOXO activation and its potential regulation of SOX2, we tested the consequence of siRNA-mediated knockdown of each gene family member, alone and in combination. Knockdown of FOXO6 using pooled siRNA constructs, but not the other FOXO proteins (individually or simultaneously), dramatically reduced erlotinib-mediated induction of SOX2 (*Figure 7B,C* and *Figure 7—figure supplement 2B*). The effect of FOXO6 knockdown on SOX2 was evident using multiple individual siRNAs targeting FOXO6 in both HCC827 and PC9 cells (*Figure 7D*; effect on other FOXO isoforms is shown in *Figure 7—figure supplement 2C*). Although some individual siRNAs targeting FOXO6 had off target effects on other FOXOs, direct targeting of FOXOs 1, 3a, and 4 had no effect on SOX2 expression, further supporting the specificity of the FOXO6 effect (*Figure 7B*). The effect of FOXO6 knockdown on SOX2 was not associated with any consistent effect on other aspects of EGFR signaling, although a moderate decrease in phospho-EGFR was observed in some experiments without significant differences in phospho-AKT or phospho-ERK (*Figure 7E*). Notably, FOXO6 expression was also heterogeneous and partially colocalized with SOX2 expression among populations of both untreated and treated cells (*Figure 7—figure supplement 3*). FOXO6 differs from other FOXO proteins in that even the inactive protein is localized in the nucleus (*Jacobs et al., 2003*; *van der Heide et al., 2005*). However, it shares the FOXO protein AKT-dependent inhibitory phosphorylation, whose suppression following repression of mutant EGFR signaling may in part explain the erlotinib-mediated FOXO6 activation.

## Erlotinib resistance in SOX2-expressing EGFR-mutant cells

Virtually all EGFR-mutant lung cancer cell lines established from patients who have not been treated with erlotinib are highly sensitive to this drug, although a few cells lines appear to be intrinsically resistant. The HCC2935 human lung cancer cell line is remarkable for harboring a characteristic oncogene-addicting EGFR mutation (exon 19 deletion) yet having unexplained resistance to erlotinib (*Figure 8A*), including absence of the common T790M gatekeeper mutation within EGFR and

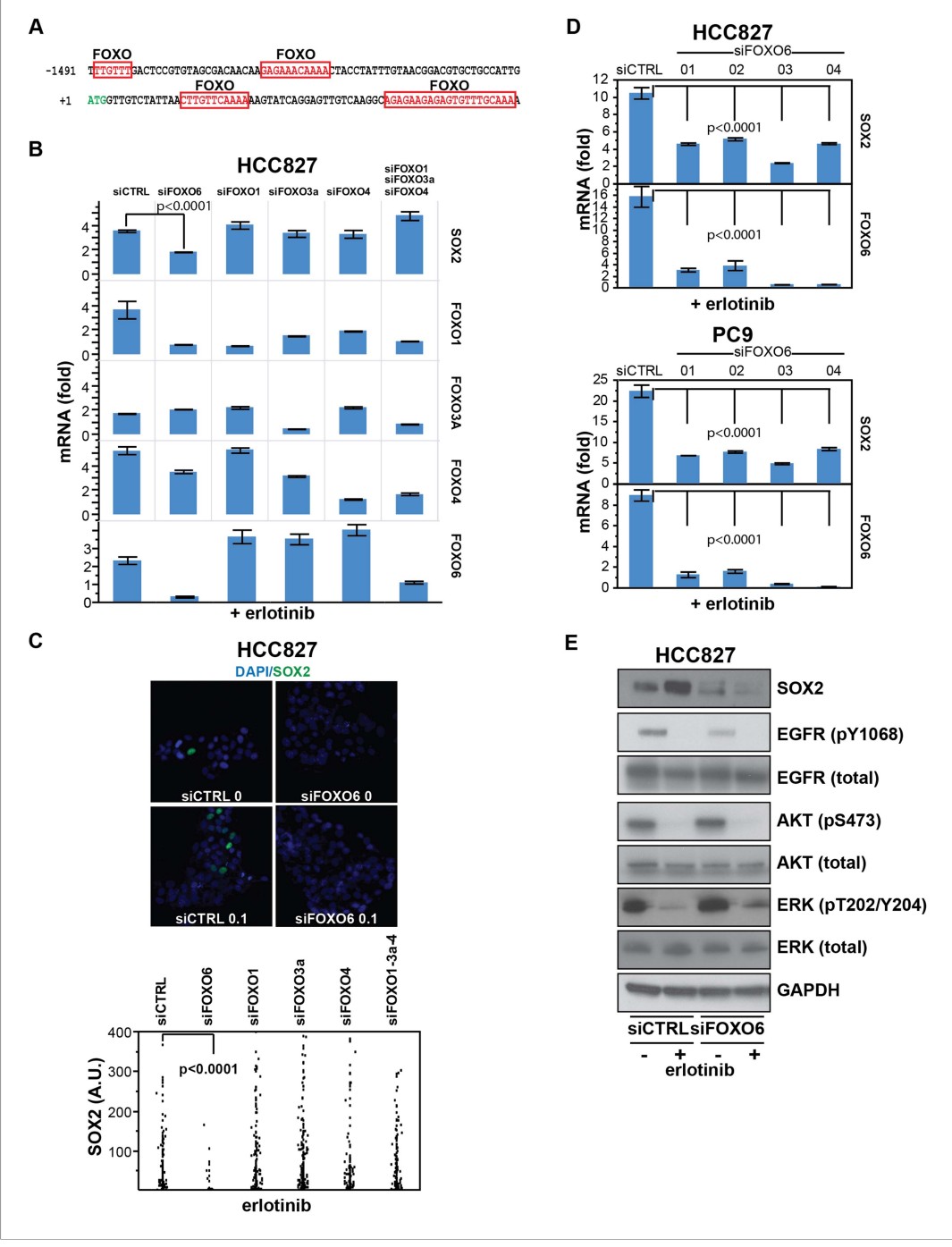

**Figure 7**. SOX2 expression in EGFR-mutant cells is regulated by FOXO6. (**A**) Putative FOXO protein binding sites within the promoter of SOX2, identified using TRANSFAC and *Zhang et al., 2011*. (**B**) HCC827 cells were transfected with control siRNA or siRNA targeting the indicated FOXO proteins (alone or FOXOs 1, 3a and 4 in combination). 72-hr after transfection, DMSO or 0.1 μM erlotinib was added for 8 hr, and the levels of SOX2 and FOXO mRNAs were determined by qPCR. Data are shown as mean Ct (normalized to GAPDH and untreated siCTRL cells) of 3 replicates −/+ SEM. Only knockdown of FOXO6 results in significantly decreased induction of SOX2 mRNA by erlotinib compared to siCTRL cells. $p < 0.0001$, other siFOXOs are without significant decrease (Student's *t*-test, unequal variance). Although siRNA pools targeting FOXOs 3a and 4 also decrease FOXO1, the lack of a SOX2 effect with specific FOXO1 knockdown argues against their role in regulation of SOX2. (**C**) The effect of FOXO6 knockdown on induction of SOX2 in HCC827 cells is shown by immunofluorescence after staining of cells with SOX2 and DAPI (upper panels) and quantitated for knockdown of all of the FOXO isoforms

*Figure 7. continued on next page*

*Figure 7. Continued*

(lower panel). Only knockdown of FOXO6 significantly decreases induction of SOX2 by erlotinib compared to siCTRL cells. p < 0.0001, other siFOXOs are without significant decrease (N = 766–1027 cells, Student's *t*-test, unequal variances). Source data are included as *Figure 7—source data 1*. Knockdown efficiency is demonstrated in *Figure 7B*. (**D**) Multiple different siRNAs effectively targeting FOXO6 block erlotinib-mediated induction of SOX2. HCC827 (upper) or PC9 (lower) cells were transfected with control siRNA or four different siRNA duplexes targeting FOXO6, treated with 0.1 μM erlotinib for 24 hr and mRNA was analyzed by qPCR. Data are shown as mean Ct (normalized to ACTB and untreated siCTRL cells) of 4 replicates −/+ SEM. p < 0.0001 for the comparison of each FOXO6 siRNA to siCTRL for both SOX2 and FOXO6 (Student's *t*-test, unequal variances). The effect of each siRNA on the levels of the other FOXO isoforms is shown in *Figure 7—figure supplement 2C*. (**E**) Knockdown of FOXO6 has minimal effects on other downstream components of EGFR signaling. HCC827 cells were transfected with siRNA targeting FOXO6 and treated with 0.1 μM erlotinib overnight followed by immunoblot analysis of protein lysates with the indicated antibodies. Knockdown of FOXO6 is demonstrated in *Figure 7—figure supplement 2A*.

The following source data and figure supplements are available for figure 7:

**Source data 1**. Raw immunofluorescence data for quantitation of SOX2 staining with different FOXO protein knockdown in *Figure 7C*.

**Source data 2**. Raw immunofluorescence data for quantitation of SOX2 and FOXO6 costaining in HCC827 cells in *Figure 7—figure supplement 3*.

**Figure supplement 1**. Recurrent FOXO binding sites in erlotinib-induced genes.

**Figure supplement 2**. FOXO6 uniquely regulates SOX2 expression.

**Figure supplement 3**. Distribution of FOXO6 vs SOX2 nuclear staining.

**Figure supplement 4**. Assessing the role of previously identified regulators on erlotinib-induced expression of SOX2.

no amplification of the MET bypass signaling pathway (*Zheng et al., 2011*). Notably, SOX2 expression at baseline is detectable in 90% of HCC2935 cells (compared to 3% of HCC827 and <1% PC9 cells), and its expression per cell is further increased upon EGFR inhibition (*Figure 8B*). To test whether increased SOX2 contributes to decreased erlotinib sensitivity in HCC2935 cells, we knocked down SOX2 using siRNA. A striking increase in erlotinib cytotoxicity was evident following SOX2 suppression (IC50 0.8 μM for siCTRL cells with no further toxicity up to 10 μM, IC50 0.1 μM for siSOX2 cells), associated with higher levels of BIM and increased PARP and caspase-3 cleavage (*Figure 8A,C*). Thus, increased baseline SOX2 contributes to erlotinib resistance in these EGFR-mutant cells.

## Discussion

The dramatic responsiveness of *EGFR*-mutated NSCLC to small molecule inhibitors of the EGFR kinase has provided a paradigm for the targeted therapy of epithelial cancers and has established a new standard of care for a genetically defined subset of patients with lung cancer. However, the invariable development of drug resistance greatly limits the effectiveness of this therapy, despite efforts to circumvent acquired genetic abnormalities, such as the recurrent T790M-EGFR gatekeeper mutation or *MET* amplification. Understanding how immediate signaling feedback loops modulate the cellular response to targeted inhibitors and how cellular heterogeneity may lead to transient drug tolerant states may thus provide important therapeutic opportunities.

### FOXO6 regulates SOX2 expression

The regulation of SOX2 expression by FOXO6, whose activation is normally repressed by EGFR signaling, is consistent with the critical role played by FOXO proteins as integrators of cellular signaling pathways. The best studied isoform, FOXO1, is highly expressed in ES cells and has been

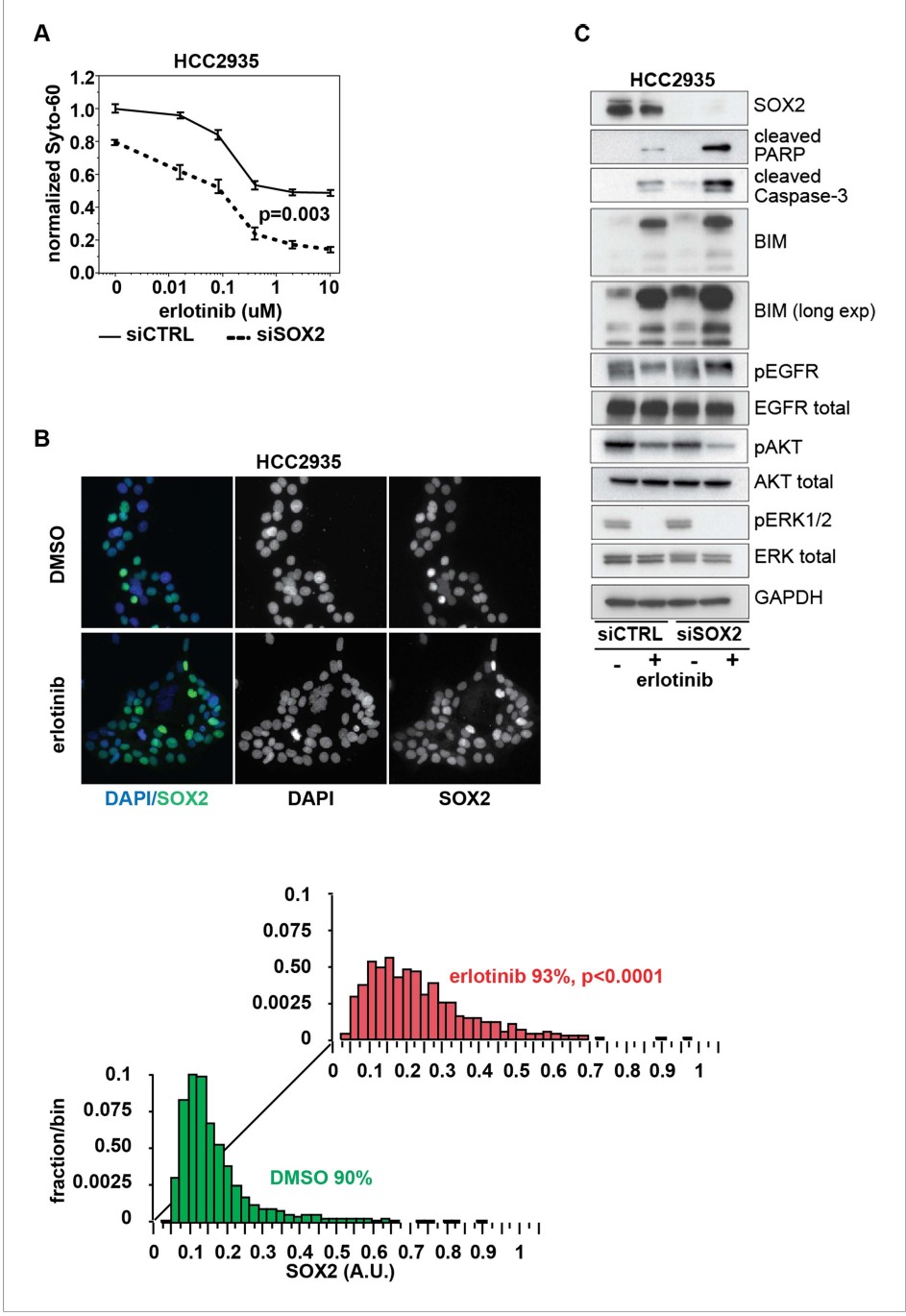

**Figure 8**. Knockdown of SOX2 sensitizes HCC2935 cells to erlotinib-induced apoptosis. (**A**) HCC2935 cells were transfected with siCTRL or siSOX2 48 hr prior to erlotinib addition and assayed for cytoxicity 48 hr later with Syto-60. Data are displayed as the mean of 5 replicates −/+ SEM. The IC50 is 0.8 µM for siCTRL and 0.1 µM for siSOX2 cells (calculated by four parameter logistic sigmoidal fit). p = 0.003 for the comparison of mean IC50 for siCTRL vs siSOX2 (Student's *t*-test, unequal variances). (**B**) Upper panels, images of untreated and erlotinib-treated HCC2935 cells, demonstrating SOX2 expression in the majority of cells. Lower panels, the distribution of SOX2 in HCC2935 was determined by quantitative immunofluorescence microscopy. p < 0.0001 for the comparison of mean SOX2 fluorescence in untreated vs treated cells (Student's *t*-test, unequal variances, N = 3342/1181, means for SOX2 fluorescence are 0.17/0.24 for untreated/treated cells, % SOX2+ is shown). Source data is included as *Figure 8—source data 1*. (**C**) HCC2935 cells were transfected with control siRNA or siRNA targeting SOX2. 48 hr after

*Figure 8. continued on next page*

*Figure 8. Continued*

transfection, DMSO or 1.0 µM erlotinib was added. The effect of SOX2 knockdown was assessed by immunoblot analysis of protein lysates with the indicated antibodies after overnight treatment.

The following source data are available for figure 8:

**Source data 1**. Raw immunofluorescence data for quantitation of SOX2 staining in HCC2935 cells in *Figure 8B*.

implicated in the maintenance of pluripotency through activation of SOX2 transcription (*Zhang et al., 2011*). All FOXO proteins bind to similar DNA sequences, with isoform-specific activity presumably conferred by cellular and promoter context (*Furuyama et al., 2000*). Indeed, all four FOXO isoforms are expressed in EGFR-mutant lung cancer cells, transcriptionally induced following EGFR inhibition and phosphorylated at homologous Serine residues, yet only FOXO6 regulates SOX2 in these cells. Given variability in knockdown efficacy (*Figure 7B*, column 6), we cannot exclude some contribution from the other FOXO family members on SOX2 expression, but FOXO6 has the dominant effect in the cells tested. Activation of FOXO6 may occur through both AKT dependent and independent pathways (*Jacobs et al., 2003*; *van der Heide et al., 2005*), and indeed we observed that treatment with PI3K inhibitors alone is insufficient for induction of SOX2 (*Figure 1B*). Other pathways that have been implicated in SOX2 regulation in the developing lung, including FGF10, WNT/beta-Catenin signaling, and TTF1 (*Que et al., 2007*; *Gontan et al., 2008*; *Hashimoto et al., 2012*), had relatively modest effects on its induction following erlotinib treatment in EGFR-mutant cancer cells (*Figure 7—figure supplement 4*), pointing to FOXO6 as the dominant pathway in this model of oncogene-dependent signaling.

## SOX2 regulates apoptosis through BIM and BMF

Our study extends the pro-apoptotic signals implicated in withdrawal of mutant EGFR signaling to include BMF, in addition to BIM and PUMA (*Gong et al., 2007*; *Bean et al., 2013*). In contrast to the latter, BMF binds with significant affinity to a subset of BCL-2 family members (BCL-2, BCL-xL, and BCL-w) (*Chen et al., 2005*; *Kuwana et al., 2005*), yet it clearly contributes to the apoptotic response to erlotinib (*Figure 6*). The direct binding by SOX2 of the *BIM* and *BMF* genes is consistent with the ability of the four reprogramming factors (SOX2, OCT-4, KLF-4, and c-MYC) to bind to the promoters of several anti-apoptotic genes (including BMF) which are induced early during the reprogramming process (*Kim et al., 2008*; *Soufi et al., 2012*). Interestingly, control of SOX2 expression and its modulation of apoptosis may differ in EGFR-mutant lung cancer, compared with other forms of NSCLC. In a recent study of lung cancers with wild-type EGFR, the activation of EGFR upregulated SOX2, thereby decreasing apoptosis through BCL2L1, a phenomenon that was notably absent in EGFR-mutant lung cancers (*Chou et al., 2013*). A similar pathway was reported in a prostate cancer cell line, also with wild-type EGFR (*Rybak and Tang, 2013*). Thus, an EGFR-SOX2-BCL2L1 pathway may be implicated in cancer cells with wild-type EGFR, whereas the EGFR-FOXO6-SOX2-BIM/BMF pathway we describe is specific for cells that are dependent on mutant EGFR signals for their survival.

## Heterogeneity of SOX2 induction and implications for targeted therapy

The heterogeneous induction of SOX2 within a clonally derived cell population could reflect the existence of an intrinsic subpopulation with heritable traits, as recently proposed in medulloblastoma and skin carcinoma (*Boumahdi et al., 2014*; *Vanner et al., 2014*). Alternatively, it could result from stochastic variation between cells in the activity of cellular signaling pathways. We favor the latter model, given the absent coexpression of SOX2 with putative stem cell markers, the failure to enrich for SOX2 positive cells following repeated erlotinib treatments, and the regeneration of heterogeneous SOX2 inducibility following single cell cloning experiments. A stochastic cell killing model for TRAIL-induced apoptosis of HeLa cells was recently described, in which naturally occurring cell-to-cell variation in the levels or activity of upstream signaling proteins leads to differential induction of mitochondrial membrane permeability and apoptosis only in some cells (*Spencer et al., 2009*).

Critically, the ability of siRNA targeting SOX2 to substantially decrease the number and rate at which resistant subclones of EGFR-mutant cells emerge following continuous erlotinib treatment suggests that, despite its heterogeneous, transient, and stochastic expression, SOX2 contributes to the emergence of stably acquired resistance (*Figure 5E*). In these respects, our observations are reminiscent of transient drug-tolerant persister cells (DTPs), also observed following erlotinib treatment (*Sharma et al., 2010*). However, there are also significant differences between the mechanisms underlying DTPs and those observed here. DTPs emerge following longer erlotinib exposure at much higher concentrations, and they constitute a lower percentage of cells within the population. Their expression of the stem cell marker CD133 and the chromatin remodeling protein KDM5A was not reproduced by SOX2+ cells (*Figure 2—figure supplements 2, 6*). Furthermore, DTPs are detectable in many cancer cell lines following treatment with multiple cytotoxic and targeted agents, whereas induction of SOX2 appears to be strictly limited to targeted EGFR inhibition in cells addicted to mutant EGFR signaling. SOX2 induction may thus be a specific and early signaling response to EGFR withdrawal, which contributes to increased cell survival, thus enhancing the likelihood of epigenetic events leading to DTPs and ultimately to stable genetic mechanisms of acquired drug resistance. Our results with HCC2935 cells (*Figure 8*) further suggest that in a subset of cases, high basal SOX2 may even blunt the initial signaling response to EGFR inhibitors.

In summary, the rewiring of cellular signaling pathways driven by a dominant mutationally activated kinase underlies oncogene addiction in cancer, providing powerful opportunities for targeted therapy. At the same time, feedback pathways that attenuate these signaling readouts may play a major role in enhancing the development of drug resistance. The EGFR-FOXO6-SOX2 signaling pathway regulating expression of the BIM and BMF apoptotic factors thus identifies a feedback loop that may attenuate the effectiveness of anti-EGFR therapy in cancer and contribute toward the ultimate development of drug resistance (*Figure 9*). The contribution of key embryonic regulators such as SOX2 points to the conservation of critical developmental pathways in cancer cells, which modulate their response to the disruption of oncogenic signals.

## Materials and methods

### Cell lines and reagents
All cell lines were grown in RPMI (GIBCO) with 10% FBS and were obtained from the ATCC or the Massachusetts General Hospital Center for Molecular Therapeutics, which performs routine cell line authentication testing by SNP and STR analysis. Erlotinib, AZD6244, BEZ235, crizotonib, lapatinib, WZ4002 (Selleckchem, Houston, TX), and BIO (Sigma, St. Louis, MO) were dissolved in DMSO.

### Microarray analysis
HCC827 cells were plated in triplicates and treated the following day for 6 hr with DMSO, erlotinib, AZD6244, and BEZ235 (each 1 µM) or for 0, 3, 6, 12, and 24 hr with erlotinib (single samples for each time point). Total mRNA was isolated using the RNeasy Mini Kit (Qiagen). Generation of cRNA, hybridization to GeneChip Human Genome U133 Plus 2.0 mRNA expression arrays and array scanning were done according to the manufacturer's standard protocols (Affymetrix, Inc.). Raw Affymetrix CEL files were converted to a single value for each probe set using Robust Multi-array Average (RMA) and normalized using quantile normalization. Quality control was performed using the distribution analysis, correlation and principle variance components analysis functions in Jmp Genomics (SAS Institute). Individual genes with statistically significantly altered expression after treatment (compared to untreated cells) were identified using ANOVA after model adjustment for multiple hypothesis testing across LSMeans differences using the False Discovery Method of Benjamini and Hochberg (FDR) with Alpha set to 0.05. The complete data set is available at the NCBI Gene Expression Omnibus (http://www.ncbi.nlm.nih.gov/geo/), Accession GSE51212.

### Quantitative PCR
For RT-qPCR, 1 µg mRNA was converted to cDNA using the First-strand cDNA Synthesis Kit (GE Healthcare, Pittsburgh, PA). cDNA was analyzed on a 7500 Real Time PCR System (Applied Biosystems) using TaqMan Gene Expression Master Mix and TaqMan gene expression assays (with

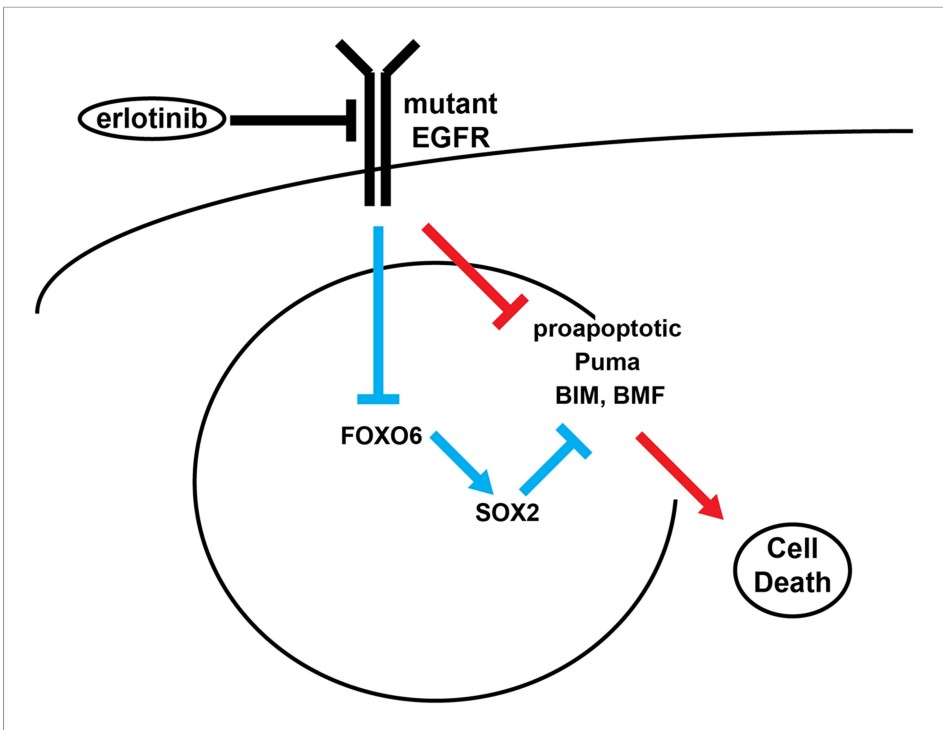

**Figure 9.** Model of SOX2 feedback signaling pathway. In untreated cells, mutant EGFR drives cell survival by activating downstream signaling pathways, including PI3K and MAPK, which inhibit apoptosis through transcriptional and post-transcriptional effects on BH3-domain proteins, including pro-apoptotic BIM and BMF. In most cells (red lines), erlotinib treatment results in EGFR inhibition, inhibition of downstream signaling and increased pro-apoptotic proteins, leading to apoptosis. The high SOX2 induced by erlotinib through activation of FOXO6 in some cells (blue lines) counteracts the pro-apoptotic effects of EGFR inhibition, sufficiently decreasing the levels of BIM and BMF to delay the apoptotic response.

GAPDH or ACTB as control, individual assays listed in *Supplementary file 1*—Thermo Fisher Scientific, Grand Island, NY). For ChIP-qPCR, immunoprecipitated chromatin (see below) was analyzed on the same system, using paired DNA PCR primers (*Supplementary file 1*) and Power SYBR Green PCR Master Mix (Applied Biosystems).

## Immunoblot analysis and antibodies

Immunoblotting was performed using standard methods. After treatment with the indicated drugs, cells were washed with cold PBS and lysed in buffer containing 20 mM Tris pH 7.5, 150 mM NaCl, 100 mM $MgCl_2$, 1% Nonidet P-40 and 10% glycerol supplemented with HALT protease and phosphatase inhibitor cocktail (Thermo Fisher Scientific) using a Q800R sonicator (Qsonica, Newtown, CT). Lysates were centrifuged at 16,000×g for 5 min at 4˚C. Protein concentrations were determined by BCA assay (Thermo Fisher Scientific). Proteins were resolved by SDS-PAGE and transferred to a polyvinylidenes difluoride membranes (Biorad) using the Transblot Turbo Transfer System (Biorad, Hercules, CA). Immunoblotting was performed per each antibody manufacturer's specifications. Antibodies used were pEGFR (Y1068), EGFR, pAKT (S473), AKT, pERK (T202/Y204), ERK, cleaved PARP, cleaved caspase-3, SOX2, BIM, phospho-FOXO1 (T24)/FoxO3a (T32), FOXO1, FOXO3 (Cell Signaling Technology, Beverly, MA), GAPDH (EMD Millipore, Billerica, MA), Ki67 (Epitomics, Burlingame, CA), phospho-FOXO6 (S184) (Abcam, Cambridge, MA), FOXO6 (Proteintech, Chicago, IL), CD133, GKLF, OCT4, MYC (Santa Cruz, Dallas, TX), CD44 and CD24 (BD Biosciences, San Jose, CA), and TY1 (Diagenode, Denville, NJ).

## Chromatin immunoprecipitation

ChIP assays were carried out using approximately 5–10 × 10⁶ HCC827 cells, following the procedures described previously (*Mikkelsen et al., 2007*; *Ku et al., 2008*). In brief, chromatin from formaldehyde-fixed

cells was fragmented to a size range of 200–700 bases with a Branson 250 sonifier. Solubilized chromatin was immunoprecipitated overnight with goat anti-SOX2 antibody or goat IgG as a negative control (both antibodies are from R&D Systems, Minneapolis, MN). Antibody–chromatin complexes were pulled down with protein G-Dynabeads (Thermo Fisher Scientific), washed, and then eluted. After crosslink reversal, RNase A, and proteinase K treatment, immunoprecipitated DNA was extracted with the Agencourt AMPure XP PCR Purification Kit (Beckman Coulter, Brea, CA). ChIP DNA was quantified with Qubit (Thermo Fisher Scientific).

### ChIP seq

5 ng purified DNA (immunoprecipiated chromatin and input controls) were used to prepare Illumina compatible sequencing libraries for sequencing using the MiSeq Desktop Sequencer (Illumina, San Diego, CA). ChIP seq reads were aligned to the hg19 reference genome using BWA (*Li and Durbin, 2009*). Aligned reads were extended to 200 bp to approximate fragment sizes, and then 25-bp resolution density maps were derived by counting the number of fragments overlapping each position, using IGV tools (*Robinson et al., 2011*). The density maps were normalized to 5 million reads, and IGV was used to visualize ChIP seq coverage maps (*Thorvaldsdottir et al., 2013*).

### Quantitative immunofluorescence analysis/immunohistochemistry

For immunofluorescence (IF) analysis, cells plated in chamber slides were fixed with 4% formaldehyde, permeabilized/blocked with 5% normal goat serum/0.3% Triton X-100 and then incubated overnight at 4°C with antibody to SOX2 (either rabbit—Cell Signaling Technology or goat—R&D systems). SOX2 staining was visualized with appropriate Alexa Fluor conjugated secondary antibodies (Jackson Immunoresearch, West Grove, PA). Ki-67 costaining was performed with antibody to Ki-67 conjugated to Alexa Fluor 488 (Epitomics). Nuclei were visualized with DAPI. Immunohistochemistry (IHC) of SOX2 on formalin-fixed, paraffin embedded tumor tissue was performed according to the antibody manufacturer's suggested protocol by the Specialized Histopathology Laboratory of the Massachusetts General Hospital using the SignalStain Boost IHC detection reagent (Cell Signaling Technology). All IF/IHC samples were imaged and quantitated using the Vectra Automated Multispectral Imaging System (PerkinElmer, Waltham, MA). Images were initially scanned at 4× magnification and then multiple high-powered fields were automatically acquired. The emission spectra of each fluorophore/IHC stain was computationally unmixed by preparing matched single stained samples. Unmixed images were segmented to identify individual cell nuclei based on DAPI or hematoxylin signal, and the mean nuclear signal for each fluorophore/IHC stain was calculated using inForm Advanced Image Analysis Software (PerkinElmer). For some samples, in addition to analyzing the mean signal for each stain on every cell analyzed, individual nuclei were scored as positive or negative for SOX2 using the scoring function in inForm to set a fixed threshold for SOX2 signal based on the presence or absence of any degree of SOX2 staining. For fluorescence images, when samples to be compared were acquired using different exposure times, data were normalized to exposure time using the normalized counts setting/function in inForm.

### Toxicity assays

Cells were plated in multiple replicate wells at 2500 cells per well in 96-well format, treated 24 hr after plating and analyzed 72 hr later. Wells were fixed with 4% formaldehyde and stained with Syto60 red fluorescence nucleic acid stain (Thermo Fisher Scientific) for 1 hr at room temperature. After washing each well three times with water, the fluorescence of each well was analyzed using a Spectramax M5 plate reader (Molecular Devices, Sunnyvale, CA) with excitation 630 nm, emission 697 nm, and cutoff 695 nm, background corrected by subtracting the mean signal from empty wells and normalized to the mean value of untreated wells.

### siRNA

All Dharmacon ON-TARGET plus siRNA pools were purchased from Thermo Fisher Scientific. Sequences are included in *Supplementary file 1*. Cells were plated at 12,500–25,000 per ml in 96-well plates (0.2 ml), 8-chamber slides (0.5 ml), 6-well plates (3 ml), or 60 mm plates (6 ml) in antibiotic-free medium and transfected the following day with each siRNA (12.5 nM final concentration) with the Dharmafect I transfection reagent (2 μl/200 μl for PC9 and 4 μl/200 μl for HCC827 and HCC2935) according to the manufacturer's standard protocol. Media were changed 6–24 hr after transfection,

and isolation of mRNA and protein lysates and immunofluorescence analysis was carried out 48–96 hr after transfection (and treatment with erlotinib).

## Lentiviral-inducible expression

Lentiviral-inducible expression constructs containing SOX2 (wild type and epitope tagged) or WNT pathway mutants under the control of a doxycycline-inducible promoter or control vector containing the GUS gene were constructed by subcloning each ORF into the pInducer 20 lentiviral vector (*Meerbrey et al., 2011*) using the Gateway cloning system (Thermo Fisher Scientific). This vector also contains a neomycin resistance cassette. Lentiviral expression constructs were cotransfected into 293T cells with the HIV-1 packaging construct pCMVdeltaR8.91 and the VSV-G envelope construct pMD.G using TransIT-LT-1 transfection reagent (Thermo Fisher Scientific). Viral supernatants were collected at 48 and 72 hr post transfection in DMEM with 30% FCS, filtered through 0.45 µm syringe filters to remove cell debris and stored in aliquots at −80°C. Transductions were carried out in the presence of 8 µg/ml Polybrene (Sigma) by spinoculation at 1200×g and at 32°C for 60 min in a Sorvall Legend RT table-top centrifuge. Viral supernatant was exchanged for fresh media 24 hr after spinoculation. To generate stable cDNA expressing cell lines, cells were selected in G418. Tetracycline-reduced FBS (Clontech, Mountain View, CA) was substituted for all media for cells transduced with the pInducer 20 vectors. To induce expression of SOX2 or WNT pathway constructs, doxycycline (Sigma) at 0.1 µg/ml was added to all cultures at least 3 hr prior to addition of erlotinib.

## Luciferase reporter assay

Stable cell lines expressing the TOP-FLASH reporter were generated by transducing cells with 7TFP recombinant lentiviruses (*Fuerer and Nusse, 2010*), and luciferase assay was performed as previously described (*Singh et al., 2012*). Briefly, Rediject D-Luciferin Ultra (Perkin Elmer) was added in 0.2 ml fresh media (1–200 dilution) to each well of cells in a 96-well plate and incubated for 15 min at 37°C. Luciferase activity was imaged with the IVIS Lumina II In Vivo Imaging System (Perkin Elmer). The radiance of each well was determined using Living Image 4.2 software (Perkin Elmer), background corrected by subtracting the mean signal from empty wells and normalized both to the relative cell number of each well as determined by Syto60 assay and the resulting normalized mean value of untreated wells.

## Xenograft tumor studies

All mouse studies were carried out according to Institutional IUCAC guidelines. Mice were treated by oral gavage with a single 100 mg/kg dose of erlotinib when subcutaneous tumors had reached ~500 mm³ in sizes (~21–28 days). PC9 and HCC827 xenograft tumors were harvested 21 hr after erlotinib treatment.

## Statistical analyses

All statistical analyses, including number of replicates and statistical method used, are included in the relevant figure legends.

## Acknowledgements

The authors wish to thank Doug Robinson (JMP Life Sciences) for help with array analysis, Charles Vanderburg (MGH Department of Neurology) for technical help and Matthew J Niederst for providing patient-derived cell lines. This work was supported by NIH RO1 CA207186 (DAH), HHMI (DAH, MNR), National Institute of Health/National Institute of Dental & Craniofacial Research K08DE020139 (SMR) and the Burroughs Wellcome Fund (MNR).

## Additional information

### Funding

| Funder | Grant reference number | Author |
| --- | --- | --- |
| Howard Hughes Medical Institute | | Daniel A Haber |
| National Institutes of Health | RO1 CA207186 | Daniel A Haber |

| Funder | Grant reference number | Author |
|---|---|---|
| National Institute of Dental and Craniofacial Research (NIDCR) | K08DE020139 | S Michael Rothenberg |

The funders had no role in study design, data collection and interpretation, or the decision to submit the work for publication.

### Author contributions
SMR, DAH, Conception and design, Acquisition of data, Analysis and interpretation of data, Drafting or revising the article, Contributed unpublished essential data or reagents; KC, SC, Conception and design, Acquisition of data, Analysis and interpretation of data; GB, Acquisition of data, Analysis and interpretation of data; ABT, Conception and design, Acquisition of data; ACF, ELL, Conception and design, Analysis and interpretation of data, Contributed unpublished essential data or reagents; MNR, Analysis and interpretation of data, Drafting or revising the article; JAE, Conception and design, Contributed unpublished essential data or reagents; SM, Conception and design, Analysis and interpretation of data, Drafting or revising the article

### Ethics
Animal experimentation: All animal studies were conducted through Institutional Animal Care and Use Committee (IUCAC)-approved animal protocol 2010N000006 from the Massachusetts General Hospital. Mice were maintained in laminar flow units in aseptic condition and the care and treatment of all mice was in in accordance with institutional guidelines.

## Additional files

### Supplementary file
• Supplementary file 1. siRNA, primer, and probe sequences/sources used in the study.

### Major dataset
The following dataset was generated:

| Author(s) | Year | Dataset title | Dataset ID and/or URL | Database, license, and accessibility information |
|---|---|---|---|---|
| Rothenberg S Michael | 2015 | Whole transcriptome analysis of erlotinib treatment in EGFR-mutant cells | GSE51212; | Publically available at the NCBI Gene Expression Omnibus (http://www.ncbi.nlm.nih.gov/geo/). |

Standard used to collect data: MIAME checklist.

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
