## [Decision Letter]

Thank you for choosing to send your work entitled “Inhibition of mutant EGFR in
lung cancer cells triggers SOX2-FOXO6 dependent survival pathways” for
consideration at *eLife*. Your full submission has been evaluated by
Charles Sawyers (Senior editor) and 3 peer reviewers, one of whom is a member of our
Board of Reviewing Editors, and the decision was reached after discussions between the
reviewers.

The comments of the reviewers are listed below. A critical issue noted by the reviewers
was that your study fails to make a direct connection between SOX2 and drug resistance.
As a result, we are returning your manuscript. The current manuscript will not be
considered further for publication. However, if new data are obtained that directly
address this criticism we would be pleased to consider a new manuscript for publication.
Please note that a new submission would be subject to full editorial review.

Targeted cancer therapies can lead to the development of drug resistance. The mechanisms
that cause drug resistance need to be understood if we are to improve the efficacy of
cancer treatment. This study, which seeks to understand resistance to EGFR-targeted
therapy, is therefore significant. The authors explored gene expression changes induced
by erlotinib that might contribute to the development of resistance. The authors report
that a FOXO6 - Sox2 pathway represses expression of the pro-apoptotic proteins BIM and
BMF to in a sub-set of cells confer survival. The authors propose that this survival
pathway provides the tumor cells with the opportunity to accumulate mutations that drive
drug resistance. This conclusion, which is not tested, is interesting and potentially
important for the design of cancer therapies.

Specific points:

1) The authors do not establish whether their observations regarding SOX2 are relevant
to any important clinical observation or outcomes. The paper is introduced with a
discussion of resistance, but the reported study does not address resistance. Does
modulation of SOX2 have effects on drug resistance?

2) To further relate the findings to drug resistance, the authors should determine what
occurs following a repeat challenge with erlotinib. The authors show that SOX2 induction
is transient. Does the proportion of cells that can increase SOX2 following erlotinib
exposure increase in the surviving cells (i.e. does erlotinib select for cells with this
phenotype)? If not, then these data may be describing some form “fractional
kill” or stochastic resistance, akin to that shown in Spencer et al., Nature,
2009. This would also be interesting, but not really related to the clinically observed
problem of acquired resistance to erlotinib and other similar inhibitors.

3) The data presented show that SOX2 expression is not only heterogeneous, but also very
rare. In Figure 2, for example, it appears that
< 20% (maybe less than 10%) of cells express SOX2 after erlotinib exposure. Thus,
it is unclear why removing SOX2 from ∼10-20% of cells would have the pronounced
effects on erlotinib-mediated death that are shown in Figure 4. The dose of erlotinib used in Figure 2 is very low. Will the proportion of cells with SOX2 induction
increase as a function of erlotinib dose?

4) At least 70% of patients who acquire resistance to EGFR inhibition do so on the basis
of identifiable new mutations in EGFR or MET. I know the authors discuss a sort-of
resistant state that SOX2. The authors should examine whether modulation of SOX2 affects
selection for mutants in vivo.

5) BIM is identified as a key player in SOX2 signaling based on SOX2 over-expression
assays (Figure 5). However, SOX2 siRNA causes
very small effects on BIM protein expression (Figure 7). Since erlotinib causes major loss of ERK activity (Figure 7), this would be anticipated to cause BIM stabilization.
One interpretation of Figure 7 is therefore that
SOX2 is not very important for BIM protein expression in this system. Figure 4 using SOX2 over-expression would appear to
support this conclusion (although in that experiment, it is not clear why SOX2 is not
induced by erlotinib in the control cells, but oddly is induced in the SOX2
over-expressing cells, which appear not to express SOX2 at time = 0).

6) BIM and BMF are identified as SOX2 target genes. Evidence to support this conclusion
is required (e.g. ChIP assays).

7) The conclusion that FOXO6, but not other FOXO family members, mediates effects of
erlotinib on SOX2 expression is not fully established. First, redundancy of FOXO
isoforms is not considered, this is especially a problem since the efficiency of
FOXO1/3a/4 knockdown is not presented. Second, ChIP assays are required to test the
chromatin interactions of these FOXO proteins in the cells employed in this study. The
connection to FOXO6 would be strengthened if it were shown that FOXO6 function was
heterogeneous and matched SOX2 expression.

8) There are a number of technical concerns with the analysis presented. First, the
siRNA studies are poorly described, for example, the siRNA are not defined. Second,
important controls are missing. Only one siRNA is presented for each knockdown and no
rescue studies were performed. Evidence that the siRNA caused a knockdown is missing;
for example in Figure 3. In other cases the
extent of knockdown appears limited (e.g. FOXO6 in Figure 3). Since the study is highly dependent on the use of siRNA
approaches, these questions limit the impact of the study.

[Editors' note: further revisions were requested prior to acceptance, as
described below.]

Thank you for choosing to send your work, “Inhibition of mutant EGFR in lung
cancer cells triggers SOX2-FOXO6 dependent survival pathways”, for further
consideration by *eLife*. Your submission has been assessed by Detlef
Weigel and Charles Sawyers in consultation with a member of the Board of Reviewing
Editors. We consider that the manuscript has been improved by the inclusion of new data
and revisions to the text. However, there are a number of points in the original review
that were not adequately addressed, specifically:

5) Effect of SOX2 on BIM protein expression. Figure 4 (now Figure 5—figure supplement 1) shows that the large increase in SOX2 expression does not appear to cause
major changes in BIM expression. Moreover, why is BIM expression in lane 5 slightly
greater than in lane 1? These data do not make a strong case for a role of SOX2 in the
regulation of BIM protein expression. This is consistent with the modest effects of SOX2
knockdown to increase BIM protein expression (Figure 5—figure supplement 4). The authors need to clarify the experimental
basis for their conclusion that BIM protein contributes to the effects of SOX2.

7) The conclusion that FOXO6 mediates the effects of erlotinib on SOX2 expression is not
well supported by the data, because it appears that the RNAi data presented is
confounded by off-target effects. Thus, siFOXO6 causes decreased expression of all other
FOXO mRNA tested (Figure 7) and siFOXO1/3a/4
causes decreased FOXO6 mRNA expression (Figure 7). These data do not allow the authors to distinguish between the roles of
different FOXO proteins (and raise questions concerning what other genes might be
affected). The authors do present additional data to address this point (Figure 7) by using more than the single siRNAs
employed elsewhere; but no controls for off-target effects on other FOXO proteins are
presented.

---

## [Author Response]

1) The authors do not establish whether their observations regarding SOX2 are
relevant to any important clinical observation or outcomes. The paper is introduced
with a discussion of resistance, but the reported study does not address resistance.
Does modulation of SOX2 have effects on drug resistance?

We now show new data in Figure 5, demonstrating
that transient siRNA-mediated knockdown of SOX2, coincident with erlotinib exposure,
decreases the subsequent development of stable, acquired resistance to erlotinib. While
continuous culture of control PC9 cells in erlotinib for two weeks leads to the
emergence of erlotinib-resistant clones, the number of resistant colonies is
significantly decreased when SOX2 induction by erlotinib is prevented with siRNA.

*2) To further relate the findings to drug resistance, the authors should
determine what occurs following a repeat challenge with erlotinib. The authors show
that SOX2 induction is transient. Does the proportion of cells that can increase SOX2
following erlotinib exposure increase in the surviving cells (i.e. does erlotinib
select for cells with this phenotype)? If not, then these data may be describing some
form “fractional kill” or stochastic resistance, akin to that shown in
Spencer et al., Nature, 2009. This would also be interesting, but not really related
to the clinically observed problem of acquired resistance to erlotinib and other
similar inhibitors*.

We undertook the requested retreatment experiments and we show in the new Figure 2 for HCC827 cells (and in Figure 2—figure supplement 4 for PC9
cells) that re-challenge with erlotinib after a period of recovery from two prior
treatments (75-90% cell killing for each treatment), does not increase the proportion of
cells with SOX2 inducible expression. Instead, SOX2 expression returns to basal levels
in the absence of drug, and re-treatment re-induces SOX2 to similar levels and with
similar cell distribution as with the prior treatment. These data are consistent with a
stochastic model of SOX2 induction. As noted by the reviewer, a similar stochastic cell
killing model was described in Spencer et al, Nature, 2009 with respect to TRAIL-induced
apoptosis. However, this in no way suggests absence of clinical significance, since (as
shown in Point #1 above), the transient expression of SOX2 itself enhances the
subsequent emergence of stably resistant colonies. Progressively increasing fractions of
SOX2 inducible cells following each erlotinib treatment are not expected in this
model.

We also now show in the new Figure 2—figure supplement 4, that in PC9 cells that have developed acquired resistance
following *long-term*, continuous erlotinib treatment, stochastic
induction of SOX2 still occurs but only at the much higher doses of erlotinib required
to inhibit EGFR in these resistant cells.

Taken together, these data support the heterogeneous, transient and stochastic induction
of SOX2 following erlotinib treatment as a precursor to the development of stable
acquired drug resistance.

*3) The data presented show that SOX2 expression is not only heterogeneous, but
also very rare. In*
Figure 2*, for example,
it appears that < 20% (maybe less than 10%) of cells express SOX2 after
erlotinib exposure. Thus, it is unclear why removing SOX2 from ∼10-20% of
cells would have the pronounced effects on erlotinib-mediated death that are shown
in*
Figure 4*. The dose of
erlotinib used in*
Figure 2
*is very low. Will the proportion of cells with SOX2 induction increase as a
function of erlotinib dose?*

In our experiments, the fraction of SOX2+ cells varies from 17-24% for HCC827
cells and 18-32% for PC9 cells (see Figure 2 and
Figure 2—figure supplement 1). Given
the threshold for immunofluorescence detection, it is likely that our immunofluorescence
assay *underestimates* the number of cells with SOX2, and as such we
propose a heterogeneous range of expression, rather than a truly rare and unique subset
of the population. It is also difficult to determine precisely what level of SOX2
expression in an individual cell is required for a measurable effect on cell survival.
We note that unlike the single cell data in Figure 2, the new Figure 5 shows an
immunoblot assay using total cell lysates, which obscures the contribution of individual
cells to the apoptotic effect of SOX2 knockdown.

As suggested by the reviewers, we also increased the dose of erlotinib beyond that
initially tested. We show in the new Figure 2—figure supplement 1, that further increasing the dose of erlotinib
beyond that required for full inhibition of EGFR signaling to 1.0μM does not
substantially increase the fraction of SOX2+ HCC827 cells (Increased cell death
in PC9 cells at the highest concentration precludes acute SOX2 induction analysis).

*4) At least 70% of patients who acquire resistance to EGFR inhibition do so on
the basis of identifiable new mutations in EGFR or MET. I know the authors discuss a
sort-of resistant state that SOX2. The authors should examine whether modulation of
SOX2 affects selection for mutants in vivo*.

The reviewers bring up the important point that stably acquired resistance to EGFR
inhibitors can take many forms, from secondary gatekeeper mutations within EGFR, to
induction of MET or other “bypass tracks”, to epithelial-mesenchymal
transition (EMT) and even lineage switching from non-small cell to small cell histology.
All of these complex mechanisms are noted in patients after prolonged treatment with
erlotinib, but only few of these are well recapitulated using the standard cell line
models, which tend to favor the gatekeeper mutation mechanism (even MET amplification is
very rare in vitro). There are ample previous publications in this area. Given its
complexity, we feel that defining the precise mechanism of long-term resistance in
various PC9 and HCC827 clones, and the extent to which the various frequencies might be
altered in the rare colonies that do ultimately emerge despite SOX2 suppression, would
be beyond the scope of this manuscript.

*5) BIM is identified as a key player in SOX2 signaling based on SOX2
over-expression assays (*Figure 5*). However, SOX2 siRNA causes very small effects on BIM
protein expression (*Figure 7*). Since erlotinib causes major loss of ERK activity
(*Figure 7*),
this would be anticipated to cause BIM stabilization. One interpretation of*
Figure 7
*is therefore that SOX2 is not very important for BIM protein expression in this
system.*
Figure 4
*using SOX2 over-expression would appear to support this conclusion (although in
that experiment, it is not clear why SOX2 is not induced by erlotinib in the control
cells, but oddly is induced in the SOX2 over-expressing cells, which appear not to
express SOX2 at time = 0)*.

The reviewers correctly point out that the activity of BIM is controlled at several
levels—both transcriptional and post-translational (in the latter case, primarily
by ERK-dependent phosphorylation/stabilization). Our data suggest transcriptional
regulation by SOX2 as an additional mechanism that contributes to BIM regulation, but it
is clearly not the only one (for example, the loss of MAPK pathway activity following
erlotinib treatment likely leads to stabilization of the pool of previously transcribed
BIM, that would be unaffected by SOX2 modulation of new BIM transcription). However, the
effect of manipulating SOX2 levels on the degree of apoptosis clearly demonstrates that
in certain cellular contexts, SOX2 can have an important, functional role.

We have replaced Figure 4 with the blot in Figure 5 (previously a supplemental figure in the
original manuscript), so as to remove any confusion about how the experiment was
performed. In this experiment, doxycycline was added for two hours to induce exogenous
SOX2 and then removed prior to the addition of erlotinib for 24 hours, followed by
preparation of lysates and immunoblot with the indicated antibodies. Here, both
induction of physiologic levels of exogenous, epitope-tagged (and slower migrating) SOX2
by doxycycline, and induction of endogenous SOX2 by erlotinib, can be clearly
appreciated, still resulting in decreased apoptotic PARP and Caspase-3 cleavage. We have
moved the original Figure 4 to Figure 5—figure supplement 1 and clarified
how the experiment was done in the accompanying legend.

*6)* BIM *and* BMF *are identified as SOX2 target
genes. Evidence to support this conclusion is required (e.g. ChIP
assays)*.

As requested by the reviewers we have performed ChIP assays. In the new Figure 6, we use CHIP seq and ChIP qPCR to
demonstrate erlotinib-induced binding of endogenous SOX2 to upstream regions of both the
*BIM* and *BMF* genes.

*7) The conclusion that FOXO6, but not other FOXO family members, mediates
effects of erlotinib on SOX2 expression is not fully established. First, redundancy
of FOXO isoforms is not considered, this is especially a problem since the efficiency
of FOXO1/3a/4 knockdown is not presented. Second, ChIP assays are required to test
the chromatin interactions of these FOXO proteins in the cells employed in this
study. The connection to FOXO6 would be strengthened if it were shown that FOXO6
function was heterogeneous and matched SOX2 expression*.

The efficiency of knockdown for all FOXO siRNAs in Figure 3 is shown in the new Figure 7
using qPCR. For all FOXO family members where good quality antibodies are available, we
have also added Western blot assays (new Figure 7—figure supplement 2). We agree with the reviewer’s comment
about the redundancy of FOXO isoforms, and this is shown in multiple FOXO knockdown
experiments (Figure 7). However, the triple
knockdown of FOXO 1, FOXO3a and FOXO4 does not decrease SOX2 induction like single FOXO6
knockdown.

To further support the role of FOXO6 alone, we use four different siRNAs to confirm that
its down regulation suppresses SOX2 induction by erlotinib in both HCC827 and PC9 cells
(Figure 7).

As requested by the reviewers, we show in the new Figure 7—figure supplement 3 that FOXO6 expression is heterogeneous,
and correlated with SOX2 expression.

We were unfortunately unable to demonstrate binding of FOXO6 to the promoter of SOX2 by
chromatin immunoprecipitation, most likely due to the poor quality of available
antibodies, since ChIP seq tracks for FOXO6-bound chromatin were inadequate across the
genome.

*8) There are a number of technical concerns with the analysis presented. First,
the siRNA studies are poorly described, for example, the siRNA are not defined.
Second, important controls are missing. Only one siRNA is presented for each
knockdown and no rescue studies were performed. Evidence that the siRNA caused a
knockdown is missing; for example in*
Figure 3*. In other cases
the extent of knockdown appears limited (e.g. FOXO6 in*
Figure 3*). Since the
study is highly dependent on the use of siRNA approaches, these questions limit the
impact of the study*.

We apologize for these omissions. siRNA target sequences are now clearly defined in the
Methods sections of the manuscript. The new Figure 5 demonstrates dramatic rescue of the apoptotic effect of the most potent
siRNA targeting SOX2 by a siRNA resistant, exogenous SOX2.

Figure 7 also now demonstrates that four
independent siRNA that effectively target FOXO6 decrease erlotinib-induced induction of
SOX2 (shown above under point 7).

The degree of FOXO6 knockdown in Figure 3 at the
protein level is 50% (despite 80-90% knockdown at the mRNA level), which may be due
technical issues related to the antibodies used (see Discussion about ChIP for FOXO6
above), or to the relatively long half-life of the protein (we waited up to 96 hours
after siRNA transfection to assay for knockdown). Consistent with some residual FOXO6
protein remaining after knockdown, we observed significant, though incomplete
suppression of SOX2 induction by erlotinib (see Figure 7).

[Editors' note: further revisions were requested prior to acceptance, as
described below.]

We consider that the manuscript has been improved by the inclusion of new data
and revisions to the text. However, there are a number of points in the original
review that were not adequately addressed, specifically:

*5) Effect of SOX2 on BIM protein expression.*
Figure 4
*(now*
Figure 5—figure supplement 1*) shows that the large increase in SOX2
expression does not appear to cause major changes in BIM expression. Moreover, why is
BIM expression in lane 5 slightly greater than in lane 1? These data do not make a
strong case for a role of SOX2 in the regulation of BIM protein expression. This is
consistent with the modest effects of SOX2 knockdown to increase BIM protein
expression (*Figure 5—figure supplement 4*). The
authors need to clarify the experimental basis for their conclusion that BIM protein
contributes to the effects of SOX2*.

The critical point about BIM is its *relative induction compared to
baseline*, rather than a comparison of basal levels across two panels. As we
show quantitatively in Figure 5—figure supplement 1 (together with the relevant protein bands from the referenced
supplementary figure), the levels of each BIM isoform are clearly increased by 12 hours
of treatment with erlotinib in the absence of ectopic SOX2 (blue lines). The
erlotinib-mediated increase in BIM is clearly *suppressed* over the same
time period by the simultaneous addition of erlotinib and ectopic SOX2 induction (red
lines). The slightly increased basal BIM levels at time zero in lane 5 versus lane 1
noted by the reviewer are most likely due to experimental variation and are not relevant
to the apoptotic effects observed.

This quantitation is now included with the revised Figure 5—figure supplement 1. In addition, we in the revised Figure 5—figure supplement 1 a different
experiment using tagged, ectopic SOX2 and extend the time of treatment to 24 hours.
Here, the levels of ectopic SOX2 induction (lanes 5-6) are comparable to endogenous SOX2
induction by erlotinib (lane 2-3); decreased BIM levels are clearly apparent (the use of
a lower percentage acrylamide gel does not permit BIM L and S to be distinguished from
EL).

To address the question of large changes in SOX2 expression leading to smaller changes
in BIM expression, we would not expect a linear relationship between arbitrary
“levels” of SOX2 versus BIM. What is important here is the functional
consequence of BIM and BMF induction (and of preventing their induction by siRNA or
ectopic SOX2), which we clearly demonstrate.

As to the major data relating to SOX2 and BIM, we note the following: in our manuscript,
we claim that the transient induction of SOX2, following acute suppression of mutant
EGFR signaling, reduces induction of *two* pro-apoptotic proteins, BIM
and BMF, thereby reducing cell death and setting the stage for acquired drug resistance
to EGFR inhibitors. This is based on several lines of evidence:

(1) Potent suppression of BIM and BMF by ectopic SOX2 (Figure 5—figure supplement 1, Figure 6 and its associated figure supplement 1, together with the new data
above).

(2) Increased BIM following SOX2 knockdown in the presence of erlotinib (Figure 5—figure supplement 4).

(3) Direct, erlotinib-dependent binding of endogenous SOX2 to both BIM and BMF
regulatory regions by ChIP assays (Figure 6).

(4) The functional consequences of this SOX2-mediated effect is demonstrated by the
cooperative suppression of erlotinib-induced apoptosis, when both BIM and BMF induction
are prevented using siRNA (Figure 6).

*7) The conclusion that FOXO6 mediates the effects of erlotinib on SOX2
expression is not well supported by the data, because it appears that the RNAi data
presented is confounded by off-target effects. Thus, siFOXO6 causes decreased
expression of all other FOXO mRNA tested (*Figure 7*) and siFOXO1/3a/4 causes decreased
FOXO6 mRNA expression (*Figure 7*). These data do not allow the authors to distinguish
between the roles of different FOXO proteins (and raise questions concerning what
other genes might be affected). The authors do present additional data to address
this point (*Figure 7*) by using more than the single siRNAs employed elsewhere;
but no controls for off-target effects on other FOXO proteins are
presented*.

Our critical finding is that *only* knockdown of FOXO6 significantly
decreases SOX2 induction by erlotinib. Given the reviewer’s concern about
possible nonspecific effects of the siRNA pool targeting FOXO6, we have now added data
on different FOXO isoform levels for each of the four individual siRNAs that comprised
the FOXO6 pool (see the revised Figure 7—figure supplement 2). Two different siRNA duplexes (-01 and -02) are uniquely
targeting FOXO6, without effect on other FOXO family members. Two other siRNAs within
the pool (-03 and -04) do have some off-target effect on FOXOs1, 3a and 4. Note in the
new figure that with 24 hours of erlotinib treatment, FOXO6 is induced by erlotinib to a
much greater degree than the other FOXOs in both HCC827 and PC9 cells, hence these
off-target effects are less impressive than the on-target effect on FOXO6. This new data
suggests that the nonspecific effects of the FOXO6 siRNA pool on other FOXOs are likely
due to duplexes -03 and -04.

In addition to *two independent siRNAs* (-01, -02) being highly specific
in knocking down FOXO6 and reducing SOX2 expression, *it is key to also note that
we have tested the other FOXO family members directly for their effect on SOX2
expression (*Figure 7*)*. siRNAs that effectively knock down either
FOXO1, FOXO3a or FOXO4 have no significant effect on SOX2. Hence, the off target effects
of FOXO siRNAs -03 and -04 on other FOXOs are not responsible for suppression of SOX2
expression.

In addition to addressing the reviewers’ concern regarding siRNA off-target
effects, the role for FOXO6 is further supported by our demonstration of FOXO6
co-localization with SOX2 (Figure 7—figure supplement 3), as requested by the reviewers of our initial submission. With
respect to other pathways, we used multiple approaches to test several other (e.g.
non-FOXO) pathways implicated in SOX2 regulation, including FGF, WNT and NKX2.1 (Figure 7—figure supplement 4), and none of
these affected SOX2 induction by erlotinib. Therefore, our conclusion, that FOXO6 is the
*primary* mediator of erlotinib’s effect on SOX2 expression, is
well supported by the data.

We thank the reviewer/editors for this opportunity to address the remaining concerns
from our resubmitted manuscript. Together with the extensive additional data and edits
provided in our revised manuscript, we provide multiple lines of evidence that SOX2
induction contributes to upfront and acquired resistance to erlotinib in EGFR-mutant
lung cancer cells. These include:

(1) Enhancement of acute erlotinib-induced apoptosis and suppression of long-term
acquired resistance, when SOX2 induction is prevented with siRNA (an effect that is
completely rescued by a SOX2 construct that is resistant to siRNA targeting).

(2) Detailed analysis of the heterogeneous, transient and stochastic nature of SOX2
induction in multiple EGFR-mutant cell lines, xenografts and patient-derived models.

(3) A proposed mechanism for both mutant EGFR’s regulation of SOX2 through FOXO6,
and SOX2’s regulation of apoptosis through BIM and BMF.

(4) A demonstration of the functional importance of high basal SOX2 for upfront
erlotinib resistance, in addition to acquired drug resistance.

The critical importance of acute and transient signaling feedback pathways for the
development of resistance to targeted anti-cancer therapies has been established for
several oncogene-addicted cancers (primarily mutant BRAF-driven melanoma and colorectal
cancer). We show that similar pathways are involved in EGFR*-*mutant lung
cancer. These studies, by their very nature, may be different from well-characterized
developmental roles of FOXO and SOX genes, but their adaptation in cancer cells and
their contribution to acquired resistance to signaling inhibitors is of significant
biological interest and high clinical relevance. Given the scope of our observations and
the readily addressed points raised by this secondary review, we are most grateful for
your consideration.